# Near-Optimal Deployment Efficiency in Reward-Free Reinforcement Learning with Linear Function Approximation

**Dan Qiao**
Department of Computer Science
UC Santa Barbara
Santa Barbara, CA 93106
danqiao@ucsb.edu

**Yu-Xiang Wang**
Department of Computer Science
UC Santa Barbara
Santa Barbara, CA 93106
yuxiangw@cs.ucsb.edu

## Abstract

We study the problem of deployment efficient reinforcement learning (RL) with linear function approximation under the *reward-free* exploration setting. This is a well-motivated problem because deploying new policies is costly in real-life RL applications. Under the linear MDP setting with feature dimension $d$ and planning horizon $H$, we propose a new algorithm that collects at most $\widetilde{O}(\frac{d^2 H^5}{\epsilon^2})$ trajectories within $H$ deployments to identify $\epsilon$-optimal policy for any (possibly data-dependent) choice of reward functions. To the best of our knowledge, our approach is the first to achieve optimal deployment complexity and optimal $d$ dependence in sample complexity at the same time, even if the reward is known ahead of time. Our novel techniques include an exploration-preserving policy discretization and a generalized G-optimal experiment design, which could be of independent interest. Lastly, we analyze the related problem of regret minimization in low-adaptive RL and provide information-theoretic lower bounds for switching cost and batch complexity.

## 1 Introduction

In many practical reinforcement learning (RL) based tasks, limited computing resources hinder applications of fully adaptive algorithms that frequently deploy new exploration policy. Instead, it is usually cheaper to collect data in large batches using the current policy deployment. Take recommendation system [Afsar et al., 2021] as an instance, the system is able to gather plentiful new data in very short time, while the deployment of a new policy often takes longer time, as it requires extensive computing and human resources. Therefore, it is impractical to switch the policy based on instantaneous data as a typical RL algorithm would demand. A feasible alternative is to run a large batch of experiments in parallel and only decide whether to update the policy after the whole batch is complete. The same constraint also appears in other RL applications such as healthcare [Yu et al., 2021], robotics [Kober et al., 2013] and new material design [Zhou et al., 2019]. In those scenarios, the agent needs to minimize the number of policy deployment while learning a good policy using (nearly) the same number of trajectories as its fully-adaptive counterparts. On the empirical side, Matsushima et al. [2020] first proposed the notion *deployment efficiency*. Later, Huang et al. [2022] formally defined *deployment complexity*. Briefly speaking, deployment complexity measures the number of policy deployments while requiring each deployment to have similar size. We measure the adaptivity of our algorithms via deployment complexity and leave its formal definition to Section 2.

Under the purpose of deployment efficiency, the recent work by Qiao et al. [2022] designed an algorithm that could solve reward-free exploration in $O(H)$ deployments. However, their sam-

Offline Reinforcement Learning Workshop at Neural Information Processing Systems, 2022.

| Algorithms for reward-free RL | Sample complexity | Deployment complexity |
|---|---|---|
| Algorithm 1 & 2 in Wang et al. [2020] | $\widetilde{O}(\frac{d^3 H^6}{\epsilon^2})$ | $\widetilde{O}(\frac{d^3 H^6}{\epsilon^2})$ |
| FRANCIS [Zanette et al., 2020b][‡] | $\widetilde{O}(\frac{d^3 H^5}{\epsilon^2})$ | $\widetilde{O}(\frac{d^3 H^5}{\epsilon^2})$ |
| RFLIN [Wagenmaker et al., 2022b][‡] | $\widetilde{O}(\frac{d^2 H^5}{\epsilon^2})$ | $\widetilde{O}(\frac{d^2 H^5}{\epsilon^2})$ |
| Algorithm 2 & 4 in Huang et al. [2022][‡] | $\widetilde{O}(\frac{d^3 H^5}{\epsilon^2 \nu_{\min}^2})^*$ | $H$ |
| LARFE [Qiao et al., 2022][†] | $\widetilde{O}(\frac{S^2 A H^5}{\epsilon^2})$ | $2H$ |
| Our Algorithm 1 & 2 (Theorem 5.1)[‡] | $\widetilde{O}(\frac{d^2 H^5}{\epsilon^2})$ | $H$ |
| Our Algorithm 1 & 2 (Theorem 7.1)[★] | $\widetilde{O}(\frac{S^2 A H^5}{\epsilon^2})$ | $H$ |
| Lower bound [Wagenmaker et al., 2022b] | $\Omega(\frac{d^2 H^2}{\epsilon^2})$ | N.A. |
| Lower bound [Huang et al., 2022] | If polynomial sample | $\widetilde{\Omega}(H)$ |

Table 1: Comparison of our results (in blue) to existing work regarding sample complexity and deployment complexity. We highlight that our results match the best known results for both sample complexity and deployment complexity at the same time. [‡]: We ignore the lower order terms in sample complexity for simplicity. [*]: $\nu_{min}$ is the problem-dependent reachability coefficient which is upper bounded by 1 and can be arbitrarily small. [†]: This work is done under tabular MDP and we transfer the $O(HSA)$ switching cost to $2H$ deployments. [★]: When both our algorithms are applied under tabular MDP, we can replace one $d$ in sample complexity by $S$.

ple complexity $\widetilde{O}(|\mathcal{S}|^2|\mathcal{A}|H^5/\epsilon^2)$, although being near-optimal under the tabular setting, can be unacceptably large under real-life applications where the state space is enormous or continuous. For environments with large state space, function approximations are necessary for representing the feature of each state. Among existing work that studies function approximation in RL, linear function approximation is arguably the simplest yet most fundamental setting. In this paper, we study deployment efficient RL with linear function approximation under the reward-free setting, and we consider the following question:

**Question 1.1.** *Is it possible to design deployment efficient and sample efficient reward-free RL algorithms with linear function approximation?*

**Our contributions.** In this paper, we answer the above question affirmatively by constructing an algorithm with near-optimal deployment and sample complexities. Our contributions are threefold.

- A new layer-by-layer type algorithm (Algorithm 1) for reward-free RL that achieves deployment complexity of $H$ and sample complexity of $\widetilde{O}(\frac{d^2 H^5}{\epsilon^2})$. Our deployment complexity is optimal while sample complexity has optimal dependence in $d$ and $\epsilon$. In addition, when applied to tabular MDP, our sample complexity (Theorem 7.1) recovers best known result $\widetilde{O}(\frac{S^2 A H^5}{\epsilon^2})$.
- We generalize G-optimal design and select near-optimal policy via uniform policy evaluation on a finite set of *representative policies* instead of using optimism and LSVI. Such technique helps tighten our sample complexity and may be of independent interest.
- We show that "No optimal-regret online learners can be deployment efficient" and deployment efficiency is incompatible with the highly relevant regret minimization setting. For regret minimization under linear MDP, we present lower bounds (Theorem 7.2 and 7.3) for other measurements of adaptivity: switching cost and batch complexity.

## 1.1 Closely related works

There is a large and growing body of literature on the statistical theory of reinforcement learning that we will not attempt to thoroughly review. Detailed comparisons with existing work on reward-free RL [Wang et al., 2020, Zanette et al., 2020b, Wagenmaker et al., 2022b, Huang et al., 2022, Qiao et al., 2022] are given in Table 1. For more discussion of relevant literature, please refer to Appendix A and the references therein. Notably, all existing algorithms under linear MDP either admit fully adaptive structure (which leads to deployment inefficiency) or suffer from sub-optimal sample complexity. In addition, when applied to tabular MDP, our algorithm has the same sample complexity and slightly better deployment complexity compared to Qiao et al. [2022].

The deployment efficient setting is slightly different from other measurements of adaptivity. The low switching setting [Bai et al., 2019] restricts the number of policy updates, while the agent can decide whether to update the policy after collecting every single trajectory. This can be difficult to implement in practical applications. A more relevant setting, the batched RL setting [Zhang et al., 2022] requires decisions about policy changes to be made at only a few (often predefined) checkpoints. Compared to batched RL, the requirement of deployment efficiency is stronger by requiring each deployment to collect the same number of trajectories. Therefore, deployment efficient algorithms are easier to deploy in parallel [see, e.g., Huang et al., 2022, for a more elaborate discussion]. Lastly, we remark that our algorithms also work under the batched RL setting by running in $H$ batches.

Technically, our method is inspired by optimal experiment design – a well-developed research area from statistics. In particular, a major technical contribution of this paper is to solve a variant of G-optimal experiment design while solving exploration in RL at the same time. Zanette et al. [2020b], Wagenmaker et al. [2022b] choose policy through *online experiment design*, i.e., running no-regret online learners to select policies adaptively for approximating the optimal design. Those online approaches, however, cannot be applied under our problem due to the requirement of deployment efficiency. To achieve deployment complexity of $H$, we can only deploy one policy for each layer, so we need to decide the policy based on sufficient exploration for only previous layers. Therefore, our approach requires *offline experiment design* and thus raises substantial technical challenge.

**A remark on technical novelty.** The general idea behind previous RL algorithms with low adaptivity is optimism and doubling schedule for updating policies that originates from UCB2 [Auer et al., 2002]. The doubling schedule, however, can not provide optimal deployment complexity. Different from those approaches, we apply layer-by-layer exploration to achieve the optimal deployment complexity, and our approach is highly non-trivial. Since we can only deploy one policy for each layer, there are two problems to be solved: the existence of a single policy that can explore all directions of a specific layer and how to find such policy. We generalize G-optimal design to show the existence of such *explorative policy*. Besides, we apply exploration-preserving *policy discretization* for approximating our generalized G-optimal design. We leave detailed discussions about these techniques to Section 3.

## 2 Problem setup

**Notations.** Throughout the paper, for $n \in \mathbb{Z}^+$, $[n] = \{1, 2, \cdots, n\}$. We denote $\|x\|_\Lambda = \sqrt{x^\top \Lambda x}$. For matrix $X \in \mathbb{R}^{d \times d}$, $\|\cdot\|_2$, $\|\cdot\|_F$, $\lambda_{\min}(\cdot)$, $\lambda_{\max}(\cdot)$ denote the operator norm, Frobenius norm, smallest eigenvalue and largest eigenvalue, respectively. For policy $\pi$, $\mathbb{E}_\pi$ and $\mathbb{P}_\pi$ denote the expectation and probability measure induced by $\pi$ under the MDP we consider. For any set $U$, $\Delta(U)$ denotes the set of all possible distributions over $U$. In addition, we use standard notations such as $O$ and $\Omega$ to absorb constants while $\widetilde{O}$ and $\widetilde{\Omega}$ suppress logarithmic factors.

**Markov Decision Processes.** We consider finite-horizon episodic *Markov Decision Processes* (MDP) with non-stationary transitions, denoted by a tuple $\mathcal{M} = (\mathcal{S}, \mathcal{A}, H, P_h, r_h)$ [Sutton and Barto, 1998], where $\mathcal{S}$ is the state space, $\mathcal{A}$ is the action space and $H$ is the horizon. The non-stationary transition kernel has the form $P_h : \mathcal{S} \times \mathcal{A} \times \mathcal{S} \mapsto [0, 1]$ with $P_h(s'|s, a)$ representing the probability of transition from state $s$, action $a$ to next state $s'$ at time step $h$. In addition, $r_h(s, a) \in \Delta([0, 1])$ denotes the corresponding distribution of reward.[1] Without loss of generality, we assume there is a fixed initial state $s_1$.[2] A policy can be seen as a series of mapping $\pi = (\pi_1, \cdots, \pi_H)$, where each $\pi_h$ maps each state $s \in \mathcal{S}$ to a probability distribution over actions, *i.e.* $\pi_h : \mathcal{S} \to \Delta(\mathcal{A})$, $\forall h \in [H]$. A random trajectory $(s_1, a_1, r_1, \cdots, s_H, a_H, r_H, s_{H+1})$ is generated by the following rule: $s_1$ is fixed, $a_h \sim \pi_h(\cdot|s_h), r_h \sim r_h(s_h, a_h), s_{h+1} \sim P_h(\cdot|s_h, a_h), \forall h \in [H]$.

**$Q$-values, Bellman (optimality) equations.** Given a policy $\pi$ and any $h \in [H]$, the value function $V_h^\pi(\cdot)$ and Q-value function $Q_h^\pi(\cdot, \cdot)$ are defined as: $V_h^\pi(s) = \mathbb{E}_\pi[\sum_{t=h}^H r_t|s_h = s], Q_h^\pi(s, a) = \mathbb{E}_\pi[\sum_{t=h}^H r_t|s_h, a_h = s, a], \forall s, a \in \mathcal{S} \times \mathcal{A}$. Besides, the value function and Q-value function with respect to the optimal policy $\pi^\star$ is denoted by $V_h^\star(\cdot)$ and $Q_h^\star(\cdot, \cdot)$. Then Bellman (optimality) equation

---

[1] We abuse the notation $r$ so that $r$ also denotes the expected (immediate) reward function.
[2] The generalized case where the initial distribution is an arbitrary distribution can be recovered from this setting by adding one layer to the MDP.

follows $\forall\, h \in [H]$:

$$Q_h^\pi(s,a) = r_h(s,a) + P_h(\cdot|s,a)V_{h+1}^\pi, \quad V_h^\pi = \mathbb{E}_{a \sim \pi_h}[Q_h^\pi],$$
$$Q_h^\star(s,a) = r_h(s,a) + P_h(\cdot|s,a)V_{h+1}^\star, \quad V_h^\star = \max_a Q_h^\star(\cdot,a).$$

In this work, we consider the reward-free RL setting, where there may be different reward functions. Therefore, we denote the value function of policy $\pi$ with respect to reward $r$ by $V^\pi(r)$. Similarly, $V^\star(r)$ denotes the optimal value under reward function $r$. We say that a policy $\pi$ is $\epsilon$-optimal with respect to $r$ if $V^\pi(r) \geq V^\star(r) - \epsilon$.

**Linear MDP [Jin et al., 2020b].** An episodic MDP $(\mathcal{S}, \mathcal{A}, H, P, r)$ is a linear MDP with known feature map $\phi : \mathcal{S} \times \mathcal{A} \to \mathbb{R}^d$ if there exist $H$ unknown signed measures $\mu_h \in \mathbb{R}^d$ over $\mathcal{S}$ and $H$ unknown reward vectors $\theta_h \in \mathbb{R}^d$ such that

$$P_h\left(s' \mid s,a\right) = \langle \phi(s,a), \mu_h\left(s'\right)\rangle, \quad r_h\left(s,a\right) = \langle \phi(s,a), \theta_h\rangle, \quad \forall\, (h,s,a,s') \in [H] \times \mathcal{S} \times \mathcal{A} \times \mathcal{S}.$$

Without loss of generality, we assume $\|\phi(s,a)\|_2 \leq 1$ for all $s,a$; and for all $h \in [H]$, $\|\mu_h(\mathcal{S})\|_2 \leq \sqrt{d}$, $\|\theta_h\|_2 \leq \sqrt{d}$.

For policy $\pi$, we define $\Lambda_{\pi,h} := \mathbb{E}_\pi[\phi(s_h, a_h)\phi(s_h, a_h)^\top]$, the *expected covariance matrix* with respect to policy $\pi$ and time step $h$ (here $s_h, a_h$ follows the distribution induced by policy $\pi$). Let $\lambda^\star = \min_{h \in [H]} \sup_\pi \lambda_{\min}(\Lambda_{\pi,h})$. We make the following assumption regarding explorability.

**Assumption 2.1** (Explorability of all directions). *The linear MDP we have satisfies $\lambda^\star > 0$.*

We remark that Assumption 2.1 only requires the existence of a (possibly non-Markovian) policy to visit all directions for each layer and it is analogous to other explorability assumptions in papers about RL under linear representation [Zanette et al., 2020b, Huang et al., 2022, Wagenmaker and Jamieson, 2022]. In addition, the parameter $\lambda^\star$ only appears in lower order terms of sample complexity bound and our algorithms do not take $\lambda^\star$ as an input.

**Reward-Free RL.** The reward-free RL setting contains two phases, the exploration phase and the planning phase. Different from PAC RL[3] setting, the learner does not observe the rewards during the exploration phase. Besides, during the planning phase, the learner has to output a near-optimal policy for any valid reward functions. More specifically, the procedure is:

1. Exploration phase: Given accuracy $\epsilon$ and failure probability $\delta$, the learner explores an MDP for $K(\epsilon, \delta)$ episodes and collects the trajectories without rewards $\{s_h^k, a_h^k\}_{(h,k) \in [H] \times [K]}$.

2. Planning phase: The learner outputs a function $\widehat{\pi}(\cdot)$ which takes reward function as input. The function $\widehat{\pi}(\cdot)$ satisfies that for any valid reward function $r$, $V^{\widehat{\pi}(r)}(r) \geq V^\star(r) - \epsilon$.

The goal of reward-free RL is to design a procedure that satisfies the above guarantee with probability at least $1 - \delta$ while collecting as few episodes as possible. According to the definition, any procedure satisfying the above guarantee is provably efficient for PAC RL setting.

**Deployment Complexity.** In this work, we measure the adaptivity of our algorithm through deployment complexity, which is defined as:

**Definition 2.2** (Deployment complexity [Huang et al., 2022]). *We say that an algorithm has deployment complexity of $M$, if the algorithm is guaranteed to finish running within $M$ deployments. In addition, the algorithm is only allowed to collect at most $N$ trajectories during each deployment, where $N$ should be fixed a priori and cannot change adaptively.*

We consider the deployment of non-Markovian policies (i.e. mixture of deterministic policies) [Huang et al., 2022]. The requirement of deployment efficiency is stronger than batched RL [Zhang et al., 2022] or low switching RL [Bai et al., 2019], which makes deployment-efficient algorithms more practical in real-life applications. For detailed comparison between these definitions, please refer to Section 1.1 and Appendix A.

## 3 Technique overview

In order to achieve the optimal deployment complexity of $H$, we apply layer-by-layer exploration. More specifically, we construct a single policy $\pi_h$ to explore layer $h$ based on previous data. Follow-

---

[3]Also known as reward-aware RL, which aims to identify near optimal policy given reward function.

ing the general methods in reward-free RL [Wang et al., 2020, Wagenmaker et al., 2022b], we do exploration through minimizing uncertainty. As will be made clear in the analysis, given exploration dataset $\mathcal{D} = \{s_h^n, a_h^n\}_{h,n \in [H] \times [N]}$, the uncertainty of layer $h$ with respect to policy $\pi$ can be characterized by $\mathbb{E}_\pi \|\phi(s_h, a_h)\|_{\Lambda_h^{-1}}$, where $\Lambda_h = I + \sum_{n=1}^N \phi(s_h^n, a_h^n)\phi(s_h^n, a_h^n)^\top$ is (regularized and unnormalized) *empirical covariance matrix*. Note that although we can not directly optimize $\Lambda_h$, we can maximize the expectation $N_{\pi_h} \cdot \mathbb{E}_{\pi_h}[\phi_h \phi_h^\top]$ ($N_{\pi_h}$ is the number of trajectories we apply $\pi_h$) by optimizing the policy $\pi_h$. Therefore, to minimize the uncertainty with respect to some policy set $\Pi$, we search for an *explorative policy* $\pi_0$ to minimize $\max_{\pi \in \Pi} \mathbb{E}_\pi \phi(s_h, a_h)(\mathbb{E}_{\pi_0} \phi_h \phi_h^\top)^{-1} \phi(s_h, a_h)$.

## 3.1 Generalized G-optimal design

For the minimization problem above, traditional G-optimal design handles the case where each deterministic policy $\pi$ generates some $\phi_\pi$ at layer $h$ with probability 1 (i.e. we directly choose $\phi$ instead of choosing $\pi$), as is the case under deterministic MDP. However, traditional G-optimal design cannot tackle our problem since under general linear MDP, each $\pi$ will generate a distribution over the feature space instead of a single feature vector. We generalize G-optimal design and show that for any policy set $\Pi$, the following Theorem 3.1 holds. More details are deferred to Appendix B.

**Theorem 3.1** (Informal version of Theorem B.1). *If there exists policy $\pi_0 \in \Delta(\Pi)$ such that $\lambda_{\min}(\mathbb{E}_{\pi_0} \phi_h \phi_h^\top) > 0$, then $\min_{\pi_0 \in \Delta(\Pi)} \max_{\pi \in \Pi} \mathbb{E}_\pi \phi(s_h, a_h)(\mathbb{E}_{\pi_0} \phi_h \phi_h^\top)^{-1} \phi(s_h, a_h) \leq d$.*

Generally speaking, Theorem 3.1 states that for any $\Pi$, there exists a single policy from $\Delta(\Pi)$ (i.e., mixture of several policies in $\Pi$) that can efficiently reduce the uncertainty with respect to $\Pi$. Therefore, assume we want to minimize the uncertainty with respect to $\Pi$ and we are able to derive the solution $\pi_0$ of the minimization above, we can simply run $\pi_0$ repeatedly for several episodes.

However, there are two gaps between Theorem 3.1 and our goal of reward free RL. First, under the Reinforcement Learning setting, the association between policy $\pi$ and the corresponding distribution of $\phi_h$ is unknown, which means we need to approximate the above minimization. It can be done by estimating the two expectations and we leave the discussion to Section 3.3. The second gap is about choosing appropriate $\Pi$ in Theorem 3.1, for which a natural idea is to use the set of all policies. It is however infeasible to simultaneously estimate the expectations for all $\pi$ accurately. The size of $\{$all policies$\}$ is infinity and $\Delta(\{$all policies$\})$ is even bigger. It seems intractable to control its complexity using existing uniform convergence techniques (e.g., a covering number argument).

## 3.2 Discretization of policy set

The key realization towards a solution to the above problem is that we do not need to consider the set of all policies. It suffices to consider a smaller subset $\Pi$ that is more amenable to an $\epsilon$-net argument. This set needs to satisfy a few conditions.

(1) Due to condition in Theorem 3.1, $\Pi$ should contain explorative policies covering all directions.

(2) $\Pi$ should contain a *representative policy set* $\Pi^{eval}$ such that it contains a near-optimal policy for any reward function.

(3) Since we apply *offline experimental design* via approximating the expectations, $\Pi$ must be "small" enough for a uniform-convergence argument to work.

We show that we can construct a finite set $\Pi$ with $|\Pi|$ being small enough while satisfying Condition (1) and (2). More specifically, given the feature map $\phi(\cdot, \cdot)$ and the desired accuracy $\epsilon$, we can construct the *explorative policy set* $\Pi_\epsilon^{exp}$ such that $\log(|\Pi_{\epsilon,h}^{exp}|) \leq \widetilde{O}(d^2 \log(1/\epsilon))$, where $\Pi_{\epsilon,h}^{exp}$ is the policy set for layer $h$. In addition, when $\epsilon$ is small compared to $\lambda^\star$, we have $\sup_{\pi \in \Delta(\Pi_\epsilon^{exp})} \lambda_{\min}(\mathbb{E}_\pi \phi_h \phi_h^\top) \geq \widetilde{\Omega}(\frac{(\lambda^\star)^2}{d})$, which verifies Condition (1).[4] Plugging in $\Pi_\epsilon^{exp}$ and approximating the minimization problem, after the exploration phase we will be able to estimate the value functions of all $\pi \in \Pi_\epsilon^{exp}$ accurately.

It remains to check Condition (2) by formalizing the *representative policy set* discussed above. From $\Pi_\epsilon^{exp}$, we can further select a subset, and we call it *policies to evaluate*: $\Pi_\epsilon^{eval}$. It satisfies that

---

[4]For more details about explorative policies, please refer to Appendix C.3.

$\log(|\Pi^{eval}_{\epsilon,h}|) = \widetilde{O}(d\log(1/\epsilon))$ while for any possible linear MDP with feature map $\phi(\cdot,\cdot)$, $\Pi^{eval}_\epsilon$ is guaranteed to contain one $\epsilon$-optimal policy. As a result, it suffices to estimate the value functions of all policies in $\Pi^{eval}_\epsilon$ and output the greedy one with the largest estimated value.[5]

## 3.3 New approach to estimate value function

Now that we have a discrete policy set, we still need to estimate the two expectations in Theorem 3.1. We design a new algorithm (Algorithm 4, details can be found in Appendix E) based on the technique of LSVI [Jin et al., 2020b] to estimate $\mathbb{E}_\pi r(s_h, a_h)$ given policy $\pi$, reward $r$ and exploration data. Algorithm 4 can estimate the expectations accurately simultaneously for all $\pi \in \Pi^{exp}$ and $r$ (that appears in the minimization problem) given sufficient exploration of the first $h - 1$ layers. Therefore, under our layer-by-layer exploration approach, after adequate exploration for the first $h - 1$ layers, Algorithm 4 provides accurate estimations for $\mathbb{E}_{\pi_0}\phi_h\phi_h^\top$ and $\mathbb{E}_\pi[\phi(s_h, a_h)^\top (\widehat{\mathbb{E}}_{\pi_0}\phi_h\phi_h^\top)^{-1}\phi(s_h, a_h)]$. As a result, the (1) we solve serves as an accurate approximation of the minimization problem in Theorem 3.1 and the solution $\pi_h$ of (1) is provably efficient in exploration.

Finally, after sufficient exploration of all $H$ layers, the last step is to estimate the value functions of all policies in $\Pi^{eval}$. We design a slightly different algorithm (Algorithm 3, details in Appendix D) for this purpose. Based on LSVI, Algorithm 3 takes $\pi \in \Pi^{eval}$ and reward function $r$ as input, and estimates $V^\pi(r)$ accurately given sufficient exploration for all $H$ layers.

# 4 Algorithms

In this section, we present our main algorithms. The algorithm for the exploration phase is Algorithm 1 which formalizes the ideas in Section 3, while the planning phase is presented in Algorithm 2.

---

**Algorithm 1** Layer-by-layer Reward-Free Exploration via Experimental Design (Exploration)

---

1: **Input:** Accuracy $\epsilon$. Failure probability $\delta$.
2: **Initialization:** $\iota = \log(dH/\epsilon\delta)$. Error budget for each layer $\bar{\epsilon} = \frac{C_1\epsilon}{H^2\sqrt{d}\cdot\iota}$. Construct $\Pi^{exp}_{\epsilon/3}$ as in Section 3.2. Number of episodes for each deployment $N = \frac{C_2 d\iota}{\bar{\epsilon}^2} = \frac{C_2 d^2 H^4 \iota^3}{C_1^2 \epsilon^2}$. Dataset $\mathcal{D} = \emptyset$.
3: **for** $h = 1, 2, \cdots, H$ **do**
4:  Solve the following optimization problem.
5: 

$$\pi_h = \underset{\pi \in \Delta(\Pi^{exp}_{\epsilon/3}) \text{ s.t. } \lambda_{\min}(\widehat{\Sigma}_\pi) \geq C_3 d^2 H\bar{\epsilon}\iota}{\operatorname{argmin}} \max_{\widehat{\pi} \in \Pi^{exp}_{\epsilon/3}} \widehat{\mathbb{E}}_{\widehat{\pi}}\left[\phi(s_h, a_h)^\top (N \cdot \widehat{\Sigma}_\pi)^{-1}\phi(s_h, a_h)\right], \quad (1)$$

6:  where $\widehat{\Sigma}_\pi$ is $\widehat{\mathbb{E}}_\pi[\phi(s_h, a_h)\phi(s_h, a_h)^\top] = \mathsf{EstimateER}(\pi, \phi(s,a)\phi(s,a)^\top, A = 1, h, \mathcal{D}, s_1)$,
7:  $\widehat{\mathbb{E}}_{\widehat{\pi}}\left[\phi(s_h, a_h)^\top (N \cdot \widehat{\Sigma}_\pi)^{-1}\phi(s_h, a_h)\right] = \mathsf{EstimateER}(\widehat{\pi}, \phi(s,a)^\top (N \cdot \widehat{\Sigma}_\pi)^{-1}\phi(s,a), A = \frac{\bar{\epsilon}}{C_2 d^3 H\iota^2}, h, \mathcal{D}, s_1)$.  // Both expectations are estimated via Algorithm 4.
8:  **for** $n = 1, 2, \cdots, N$ **do**
9:   Run $\pi_h$ and add trajectory $\{s_i^n, a_i^n\}_{i \in [H]}$ to $\mathcal{D}$.  // Run Policy $\pi_h$ for $N$ episodes.
10:  **end for**
11: **end for**
12: **Output: Dataset** $\mathcal{D}$.

---

**Exploration Phase.** We apply layer-by-layer exploration and $\pi_h$ is the stochastic policy we deploy to explore layer $h$. For solving $\pi_h$, we approximate generalized G-optimal design via (1). For each candidate $\pi$ and $\widehat{\pi}$, we estimate the two expectations by calling $\mathsf{EstimateER}$ (Algorithm 4, details in Appendix E). $\mathsf{EstimateER}$ is a generic subroutine for estimating the value function under a particular reward design. We estimate the two expectations of interest by carefully choosing one specific reward design for each coordinate separately, so that the resulting value function provides an estimate to the

---

[5]For more details about policies to evaluate, please refer to Appendix C.2.

desired quantity in that coordinate. [6] As mentioned above and will be made clear in the analysis, given adequate exploration of the first $h-1$ layers, all estimations will be accurate and the surrogate policy $\pi_h$ is sufficiently explorative for all directions at layer $h$.

The restriction on $\lambda_{\min}(\widehat{\Sigma}_\pi)$ is for technical reason only, and we will show that under the assumption in Theorem 5.1, there exists valid solution of (1). Lastly, we remark that solving (1) is inefficient in general. Detailed discussions about computation are deferred to Section 7.2.

---

**Algorithm 2** Find Near-Optimal Policy Given Reward Function (Planning)

---

1: **Input:** Dataset $\mathcal{D}$ from Algorithm 1. Feasible linear reward function $r = \{r_h\}_{h \in [H]}$.
2: **Initialization:** Construct $\Pi_{\epsilon/3}^{eval}$ as in Section 3.2.  // The set of policies to evaluate.
3: **for** $\pi \in \Pi_{\epsilon/3}^{eval}$ **do**
4:  $\widehat{V}^\pi(r) = \text{EstimateV}(\pi, r, \mathcal{D}, s_1)$.  // Estimate value functions using Algorithm 3.
5: **end for**
6: $\widehat{\pi} = \arg\max_{\pi \in \Pi_{\epsilon/3}^{eval}} \widehat{V}^\pi(r)$.  // Output the greedy policy w.r.t $\widehat{V}^\pi(r)$.
7: **Output:** Policy $\widehat{\pi}$.

---

**Planning Phase.** The output dataset $\mathcal{D}$ from the exploration phase contains sufficient information for the planning phase. In the planning phase (Algorithm 2), we construct a set of policies to evaluate and repeatedly apply Algorithm 3 (in Appendix D) to estimate the value function of each policy given reward function. Finally, Algorithm 2 outputs the policy with the highest estimated value. Since $\mathcal{D}$ has acquired sufficient information, all possible estimations in line 4 are accurate. Together with the property that there exists near-optimal policy in $\Pi_{\epsilon/3}^{eval}$, we have that the output $\widehat{\pi}$ is near-optimal.

# 5 Main results

In this section, we state our main results, which formalize the techniques and algorithmic ideas we discuss in previous sections.

**Theorem 5.1.** *We run Algorithm 1 to collect data and let Planning$(\cdot)$ denote the output of Algorithm 2. There exist universal constants $C_1, C_2, C_3, C_4 > 0$[7] such that for any accuracy $\epsilon > 0$ and failure probability $\delta > 0$, as well as $\epsilon < \frac{H(\lambda^\star)^2}{C_4 d^{7/2} \log(1/\lambda^\star)}$, with probability $1 - \delta$, for any feasible linear reward function $r$, Planning$(r)$ returns a policy that is $\epsilon$-optimal with respect to $r$. In addition, the deployment complexity of Algorithm 1 is $H$ while the number of trajectories is $\widetilde{O}(\frac{d^2 H^5}{\epsilon^2})$.*

The proof of Theorem 5.1 is sketched in Section 6 with details in the Appendix. Below we discuss some interesting aspects of our results.

**Near optimal deployment efficiency.** First, the deployment complexity of our Algorithm 1 is optimal up to a log-factor among all reward-free algorithms with polynomial sample complexity, according to a $\Omega(H/\log_d(NH))$ lower bound (Theorem B.3 of Huang et al. [2022]). In comparison, the deployment complexity of RFLIN [Wagenmaker et al., 2022b] can be the same as their sample complexity (also $\widetilde{O}(d^2 H^5/\epsilon^2)$) in the worst case.

**Near optimal sample complexity.** Secondly, our sample complexity matches the best-known sample complexity $\widetilde{O}(d^2 H^5/\epsilon^2)$ [Wagenmaker et al., 2022b] of reward-free RL even when deployment efficiency is not needed. It is also optimal in parameter $d$ and $\epsilon$ up to lower-order terms, when compared against the lower bound of $\Omega(d^2 H^2/\epsilon^2)$ (Theorem 2 of Wagenmaker et al. [2022b]).

**Dependence on $\lambda^\star$.** A striking difference of our result comparing to the closest existing work [Huang et al., 2022] is that the sample complexity is independent to the explorability parameter $\lambda^\star$

---

[6]For $\widehat{\Sigma}_\pi$, what we need to handle is matrix reward $\phi_h \phi_h^\top$ and stochastic policy $\pi \in \Delta(\Pi_{\epsilon/3}^{exp})$, we apply generalized version of Algorithm 4 to tackle this problem as discussed in Appendix F.1.

[7]$C_1, C_2, C_3$ are the universal constants in Algorithm 1.

in the small-$\epsilon$ regime. This is highly desirable because we only require a non-zero $\lambda^\star$ to exist, and smaller $\lambda^\star$ does not affect the sample complexity asymptotically. In addition, our algorithm does not take $\lambda^\star$ as an input (although we admit that the theoretical guarantee only holds when $\epsilon$ is small compared to $\lambda^\star$). In contrast, the best existing result (Algorithm 2 of Huang et al. [2022]) requires the knowledge of explorability parameter $\nu_{\min}$[8] and a sample complexity of $\widetilde{O}(1/\epsilon^2 \nu_{\min}^2)$ for any $\epsilon > 0$. We leave detailed comparisons with Huang et al. [2022] to Appendix G.

**Sample complexity in the large-$\epsilon$ regime.** For the case when $\epsilon$ is larger than the threshold: $\frac{H(\lambda^\star)^2}{C_4 d^{7/2} \log(1/\lambda^\star)}$, we can run the procedure with $\epsilon = \frac{H(\lambda^\star)^2}{C_4 d^{7/2} \log(1/\lambda^\star)}$, and the sample complexity will be $\widetilde{O}(\frac{d^9 H^3}{(\lambda^\star)^4})$. So the overall sample complexity for any $\epsilon > 0$ can be bounded by $\widetilde{O}(\frac{d^2 H^5}{\epsilon^2} + \frac{d^9 H^3}{(\lambda^\star)^4})$. This effectively says that the algorithm requires a "Burn-In" period before getting non-trivial results. Similar limitations were observed for linear MDPs before [Huang et al., 2022, Wagenmaker and Jamieson, 2022] so it is not a limitation of our analysis.

**Comparison to Qiao et al. [2022].** Algorithm 4 (LARFE) of Qiao et al. [2022] tackles reward-free exploration under tabular MDP in $O(H)$ deployments while collecting $\widetilde{O}(\frac{S^2 A H^5}{\epsilon^2})$ trajectories. We generalize their result to reward-free RL under linear MDP with the same deployment complexity. More importantly, although a naive instantiation of our main theorem to the tabular MDP only gives $\widetilde{O}(\frac{S^2 A^2 H^5}{\epsilon^2})$, a small modification to an intermediate argument gives the same $\widetilde{O}(\frac{S^2 A H^5}{\epsilon^2})$, which matches the best-known results for tabular MDP. More details will be discussed in Section 7.1.

# 6 Proof sketch

In this part, we sketch the proof of Theorem 5.1. Notations $\iota, \bar{\epsilon}, C_i \ (i \in [4]), \Pi^{exp}, \Pi^{eval}, \widehat{\Sigma}_\pi$ and $\widehat{\mathbb{E}}_\pi$ are defined in Algorithm 1. We start with the analysis of deployment complexity.

**Deployment complexity.** Since for each layer $h \in [H]$, we only deploy one stochastic policy $\pi_h$ for exploration, the deployment complexity is $H$. Next we focus on the sample complexity.

**Sample complexity.** Our proof of sample complexity bound results from induction. With the choice of $\bar{\epsilon}$ and $N$ from Algorithm 1, suppose that $\Lambda_{\tilde{h}}^k$ is empirical covariance matrix from data up to the $k$-th deployment[9], we assume $\max_{\pi \in \Pi_{\epsilon/3}^{exp}} \mathbb{E}_\pi[\sum_{\tilde{h}=1}^{h-1} \sqrt{\phi(s_{\tilde{h}}, a_{\tilde{h}})^\top (\Lambda_{\tilde{h}}^{h-1})^{-1} \phi(s_{\tilde{h}}, a_{\tilde{h}})}] \leq (h-1)\bar{\epsilon}$ holds and prove that with high probability, $\max_{\pi \in \Pi_{\epsilon/3}^{exp}} \mathbb{E}_\pi[\sqrt{\phi(s_h, a_h)^\top (\Lambda_h^h)^{-1} \phi(s_h, a_h)}] \leq \bar{\epsilon}$.

Note that the induction condition implies that the uncertainty for the first $h-1$ layers is small, we have the following key lemma that bounds the estimation error of $\widehat{\Sigma}_\pi$ from (1).

**Lemma 6.1.** *With high probability, for all $\pi \in \Delta(\Pi_{\epsilon/3}^{exp})$, $\|\widehat{\Sigma}_\pi - \mathbb{E}_\pi \phi_h \phi_h^\top\|_2 \leq \frac{C_3 d^2 H \bar{\epsilon} \iota}{4}$.*

According to our assumption on $\epsilon$, the *optimal* policy for exploration $\bar{\pi}_h^\star$ [10] satisfies that $\lambda_{\min}(\mathbb{E}_{\bar{\pi}_h^\star} \phi_h \phi_h^\top) \geq \frac{5C_3 d^2 H \bar{\epsilon} \iota}{4}$. Therefore, $\bar{\pi}_h^\star$ is a feasible solution of (1) and it holds that:

$$\max_{\widehat{\pi} \in \Pi_{\epsilon/3}^{exp}} \widehat{\mathbb{E}}_{\widehat{\pi}} \left[ \phi(s_h, a_h)^\top (N \cdot \widehat{\Sigma}_{\pi_h})^{-1} \phi(s_h, a_h) \right] \leq \max_{\widehat{\pi} \in \Pi_{\epsilon/3}^{exp}} \widehat{\mathbb{E}}_{\widehat{\pi}} \left[ \phi(s_h, a_h)^\top (N \cdot \widehat{\Sigma}_{\bar{\pi}_h^\star})^{-1} \phi(s_h, a_h) \right].$$

Moreover, due to matrix concentration and the Lemma 6.1 we derive, we can prove that $(\frac{4}{5} \Sigma_{\bar{\pi}_h^\star})^{-1} \succcurlyeq (\widehat{\Sigma}_{\bar{\pi}_h^\star})^{-1}$ and $(N \cdot \widehat{\Sigma}_{\pi_h})^{-1} \succcurlyeq (2\Lambda_h^h)^{-1}$. [11] In addition, similar to the estimation error of $\widehat{\Sigma}_\pi$, the following lemma bounds the estimation error of $\widehat{\mathbb{E}}_{\widehat{\pi}}[\phi(s_h, a_h)^\top (N \cdot \widehat{\Sigma}_\pi)^{-1} \phi(s_h, a_h)]$ from (1).

---

[8]$\nu_{\min}$ in Huang et al. [2022] is defined as $\nu_{\min} = \min_{h \in [H]} \min_{\|\theta\|=1} \max_\pi \sqrt{\mathbb{E}_\pi[(\phi_h^\top \theta)^2]}$, which is also measurement of explorability. Note that $\nu_{\min}$ is always upper bounded by 1 and can be arbitrarily small.

[9]Detailed definition is deferred to Appendix F.4.

[10]Solution of the actual minimization problem, detailed definition in (39).

[11]$\Sigma_{\bar{\pi}_h^\star} = \mathbb{E}_{\bar{\pi}_h^\star}[\phi_h \phi_h^\top]$. The proof is through direct calculation, details are deferred to Appendix F.6.

**Lemma 6.2.** *With high probability, for all $\widehat{\pi} \in \Pi_{\epsilon/3}^{exp}$, $\pi \in \Delta(\Pi_{\epsilon/3}^{exp})$ such that $\lambda_{\min}(\widehat{\Sigma}_\pi) \geq C_3 d^2 H \bar{\epsilon} \iota$,*

$$\left| \widehat{\mathbb{E}}_{\widehat{\pi}} \left[ \phi(s_h, a_h)^\top (N \cdot \widehat{\Sigma}_\pi)^{-1} \phi(s_h, a_h) \right] - \mathbb{E}_{\widehat{\pi}} \left[ \phi(s_h, a_h)^\top (N \cdot \widehat{\Sigma}_\pi)^{-1} \phi(s_h, a_h) \right] \right| \leq \frac{\bar{\epsilon}^2}{2d^2} \leq \frac{\bar{\epsilon}^2}{8}.$$

With all the conclusions above, we have ($\Sigma_\pi$ is short for $\mathbb{E}_\pi[\phi_h \phi_h^\top]$):

$$\frac{3\bar{\epsilon}^2}{8} \geq \frac{5d}{4N} + \frac{\bar{\epsilon}^2}{8} \geq \max_{\widehat{\pi} \in \Pi_{\epsilon/3}^{exp}} \mathbb{E}_{\widehat{\pi}}[\phi(s_h, a_h)^\top (\frac{4N}{5} \cdot \Sigma_{\bar{\pi}_h^\star})^{-1} \phi(s_h, a_h)] + \frac{\bar{\epsilon}^2}{8}$$

$$\geq \max_{\widehat{\pi} \in \Pi_{\epsilon/3}^{exp}} \mathbb{E}_{\widehat{\pi}}[\phi(s_h, a_h)^\top (N \cdot \widehat{\Sigma}_{\bar{\pi}_h^\star})^{-1} \phi(s_h, a_h)] + \frac{\bar{\epsilon}^2}{8}$$

$$\geq \max_{\widehat{\pi} \in \Pi_{\epsilon/3}^{exp}} \widehat{\mathbb{E}}_{\widehat{\pi}}[\phi(s_h, a_h)^\top (N \cdot \widehat{\Sigma}_{\bar{\pi}_h^\star})^{-1} \phi(s_h, a_h)] \geq \max_{\widehat{\pi} \in \Pi_{\epsilon/3}^{exp}} \widehat{\mathbb{E}}_{\widehat{\pi}}[\phi(s_h, a_h)^\top (N \cdot \widehat{\Sigma}_{\pi_h})^{-1} \phi(s_h, a_h)]$$

$$\geq \max_{\widehat{\pi} \in \Pi_{\epsilon/3}^{exp}} \mathbb{E}_{\widehat{\pi}}[\phi(s_h, a_h)^\top (N \cdot \widehat{\Sigma}_{\pi_h})^{-1} \phi(s_h, a_h)] - \frac{\bar{\epsilon}^2}{8} \geq \max_{\widehat{\pi} \in \Pi_{\epsilon/3}^{exp}} \mathbb{E}_{\widehat{\pi}}[\phi(s_h, a_h)^\top (2\Lambda_h^h)^{-1} \phi(s_h, a_h)] - \frac{\bar{\epsilon}^2}{8}$$

$$\geq \frac{1}{2} \max_{\widehat{\pi} \in \Pi_{\epsilon/3}^{exp}} \left( \mathbb{E}_{\widehat{\pi}} \sqrt{\phi(s_h, a_h)^\top (\Lambda_h^h)^{-1} \phi(s_h, a_h)} \right)^2 - \frac{\bar{\epsilon}^2}{8}.$$

As a result, the induction holds. Together with the fact that $\Pi_{\epsilon/3}^{eval}$ is subset of $\Pi_{\epsilon/3}^{exp}$, we have $\max_{\pi \in \Pi_{\epsilon/3}^{eval}} \mathbb{E}_\pi[\sum_{h=1}^H \sqrt{\phi(s_h, a_h)^\top (\Lambda_h)^{-1} \phi(s_h, a_h)}] \leq H\bar{\epsilon}$. We have the following lemma.

**Lemma 6.3.** *With high probability, for all $\pi \in \Pi_{\epsilon/3}^{eval}$ and $r$, $|\widehat{V}^\pi(r) - V^\pi(r)| \leq \widetilde{O}(H\sqrt{d}) \cdot H\bar{\epsilon} \leq \frac{\epsilon}{3}$.*

Finally, since $\Pi_{\epsilon/3}^{eval}$ contains $\epsilon/3$-optimal policy, the greedy policy with respect to $\widehat{V}^\pi(r)$ is $\epsilon$-optimal.

# 7 Some discussions

In this section, we discuss some interesting extensions of our main results.

## 7.1 Application to Tabular MDP

Under the special case where the linear MDP is actually a tabular MDP and the feature map is canonical basis [Jin et al., 2020b], our Algorithm 1 and 2 are still provably efficient. Suppose the tabular MDP has discrete state-action space with cardinality $|\mathcal{S}| = S$, $|\mathcal{A}| = A$, let $d_m = \min_h \sup_\pi \min_{s,a} d_h^\pi(s, a) > 0$ where $d_h^\pi$ is occupancy measure, then the following theorem holds.

**Theorem 7.1** (Informal version of Theorem H.2). *With minor revision to Algorithm 1 and 2, when $\epsilon$ is small compared to $d_m$, our algorithms can solve reward-free exploration under tabular MDP within $H$ deployments and the sample complexity is bounded by $\widetilde{O}(\frac{S^2 A H^5}{\epsilon^2})$.*

The detailed version and proof of Theorem 7.1 are deferred to Appendix H.1 due to space limit. We highlight that we recover the best known result from Qiao et al. [2022] under mild assumption about reachability to all (state,action) pairs. The replacement of one $d$ by $S$ is mainly because under tabular MDP, there are $A^S$ different deterministic policies for layer $h$ and the log-covering number of $\Pi_h^{eval}$ can be improved from $\widetilde{O}(d)$ to $\widetilde{O}(S)$. In this way, we effectively save a factor of $A$.

## 7.2 Computational efficiency

We admit that solving the optimization problem (1) is inefficient in general, while this can be solved approximately in exponential time by enumerating $\pi$ from a tight covering set of $\Delta(\Pi_{\epsilon/3}^{exp})$. Note that the issue of computational tractability arises in many previous works [Zanette et al., 2020a, Wagenmaker and Jamieson, 2022] that focused on information-theoretic results under linear MDP, and such issue is usually not considered as a fundamental barrier. For efficient surrogate of (1), we remark that a possible method is to apply softmax (or other differentiable) representation of the policy space and use gradient-based optimization techniques to find approximate solution of (1).

### 7.3 Possible extensions to regret minimization with low adaptivity

In this paper, we tackle the problem of deployment efficient reward-free exploration while the optimal adaptivity under regret minimization still remains open. We remark that deployment complexity is not an ideal measurement of adaptivity for this problem since the definition requires all deployments to have similar sizes, which forces the deployment complexity to be $\widetilde{\Omega}(\sqrt{T})$ if we want regret bound of order $\widetilde{O}(\sqrt{T})$. Therefore, the more reasonable task is to design algorithms with near optimal switching cost or batch complexity. We present the following two lower bounds whose proof is deferred to Appendix H.2. Here the number of episodes is $K$ and the number of steps $T := KH$.

**Theorem 7.2.** *For any algorithm with the optimal $\widetilde{O}(\sqrt{poly(d, H)T})$ regret bound, the switching cost is at least $\Omega(dH \log \log T)$.*

**Theorem 7.3.** *For any algorithm with the optimal $\widetilde{O}(\sqrt{poly(d, H)T})$ regret bound, the number of batches is at least $\Omega(\frac{H}{\log_d T} + \log \log T)$.*

To generalize our Algorithm 1 to regret minimization, what remains is to remove Assumption 2.1. Suppose we can do accurate uniform policy evaluation (as in Algorithm 2) with low adaptivity without assumption on explorability of policy set, then we can apply iterative policy elimination (i.e., eliminate the policies that are impossible to be optimal) and do exploration with the remaining policies. Although Assumption 2.1 is common in relevant literature, it is not necessary intuitively since under linear MDP, if some direction is hard to encounter, we do not necessarily need to gather much information on this direction. Under tabular MDP, Qiao et al. [2022] applied absorbing MDP to ignore those "hard to visit" states and we leave generalization of such idea as future work.

## 8 Conclusion

In this work, we studied the well-motivated deployment efficient reward-free RL with linear function approximation. Under the linear MDP model, we designed a novel reward-free exploration algorithm that collects $\widetilde{O}(\frac{d^2 H^5}{\epsilon^2})$ trajectories in only $H$ deployments. And both the sample and deployment complexities are near optimal. An interesting future direction is to design algorithms to match our lower bounds for regret minimization with low adaptivity. We believe the techniques we develop (generalized G-optimal design and exploration-preserving policy discretization) could serve as basic building blocks and we leave the generalization as future work.

## Acknowledgments

The research is partially supported by NSF Awards #2007117. The authors would like to thank Jiawei Huang and Nan Jiang for explaining the result of their paper.

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

# A Extended related works

**Low regret reinforcement learning algorithms.** Regret minimization under tabular MDP has been extensively studied by a long line of works [Brafman and Tennenholtz, 2002, Kearns and Singh, 2002, Jaksch et al., 2010, Osband et al., 2013, Agrawal and Jia, 2017, Jin et al., 2018]. Among those optimal results, Azar et al. [2017] achieved the optimal regret bound $\widetilde{O}(\sqrt{HSAT})$ for stationary MDP through model-based algorithm, while Zhang et al. [2020c] applied Q-learning type algorithm to achieve the optimal $\widetilde{O}(\sqrt{H^2SAT})$ regret under non-stationary MDP. Dann et al. [2019] provided policy certificates in addition to stating optimal regret bound. Different from these minimax optimal algorithms, Zanette and Brunskill [2019] derived problem-dependent regret bound, which can imply minimax regret bound. Another line of works studied regret minimization under linear MDP. Yang and Wang [2019] developed the first efficient algorithm for linear MDP with simulator. Jin et al. [2020b] applied LSVI-UCB to achieve the regret bound of $\widetilde{O}(\sqrt{d^3H^3T})$. Later, Zanette et al. [2020a] improved the regret bound to $\widetilde{O}(\sqrt{d^2H^3T})$ at the cost of computation. Recently, Hu et al. [2022] first reached the minimax optimal regret $\widetilde{O}(\sqrt{d^2H^2T})$ via a computationally efficient algorithm. There are some other works studying the linear mixture MDP setting [Ayoub et al., 2020, Zhou et al., 2021, Zhang et al., 2021b] or more general settings like MDP with low Bellman Eluder dimension [Jin et al., 2021].

**Reward-free exploration.** Jin et al. [2020a] first studied the problem of reward-free exploration, they designed an algorithm while using EULER [Zanette and Brunskill, 2019] for exploration and arrived at the sample complexity of $\widetilde{O}(S^2AH^5/\epsilon^2)$. This sample complexity was improved by Kaufmann et al. [2021] to $\widetilde{O}(S^2AH^4/\epsilon^2)$ by building upper confidence bound for any reward function and any policy. Finally, minimax optimal result $\widetilde{O}(S^2AH^3/\epsilon^2)$ was derived in Ménard et al. [2021] by constructing a novel exploration bonus. At the same time, a more general optimal result was achieved by Zhang et al. [2020b] who considered MDP with stationary transition kernel and uniformly bounded reward. Zhang et al. [2020a] studied a similar setting named task-agnostic exploration and designed an algorithm that can find $\epsilon$-optimal policies for $N$ arbitrary tasks after at most $\widetilde{O}(SAH^5 \log N/\epsilon^2)$ episodes. For linear MDP setting, Wang et al. [2020] generalized LSVI-UCB and arrived at the sample complexity of $\widetilde{O}(d^3H^6/\epsilon^2)$. The sample complexity was improved by Zanette et al. [2020b] to $\widetilde{O}(d^3H^5/\epsilon^2)$ through approximating G-optimal design. Recently, Wagenmaker et al. [2022b] did exploration through applying first-order regret algorithm [Wagenmaker et al., 2022a] and achieved sample complexity bound of $\widetilde{O}(d^2H^5/\epsilon^2)$, which matches their lower bound $\Omega(d^2H^2/\epsilon^2)$ up to $H$ factors. There are other reward-free works under linear mixture MDP [Chen et al., 2021, Zhang et al., 2021a]. Meanwhile, there is a new setting that aims to do reward-free exploration under low adaptivity and Huang et al. [2022], Qiao et al. [2022] designed provably efficient algorithms for linear MDP and tabular MDP, respectively.

**Low switching algorithms for bandits and RL.** There are two kinds of switching costs. Global switching cost simply measures the number of policy switches, while local switching cost is defined (only under tabular MDP) as $N_{switch}^{local} = \sum_{k=1}^{K-1} |\{(h,s) \in [H] \times \mathcal{S} : \pi_k^h(s) \neq \pi_{k+1}^h(s)\}|$ where $K$ is the number of episodes. For multi-armed bandits with $A$ arms and $T$ episodes, Cesa-Bianchi et al. [2013] first achieved the optimal $\widetilde{O}(\sqrt{AT})$ regret with only $O(A \log \log T)$ policy switches. Simchi-Levi and Xu [2019] generalized the result by showing that to get optimal $\widetilde{O}(\sqrt{T})$ regret bound, both the switching cost upper and lower bounds are of order $A \log \log T$. Under stochastic linear bandits, Abbasi-Yadkori et al. [2011] applied doubling trick to achieve the optimal regret $\widetilde{O}(d\sqrt{T})$ with $O(d \log T)$ policy switches. Under slightly different setting, Ruan et al. [2021] improved the result by improving the switching cost to $O(\log \log T)$ without worsening the regret bound. Under tabular MDP, Bai et al. [2019] applied doubling trick to Q-learning and reached regret bound $\widetilde{O}(\sqrt{H^3SAT})$ with local switching cost $O(H^3SA \log T)$. Zhang et al. [2020c] applied advantage decomposition to improve the regret bound and local switching cost bound to $\widetilde{O}(\sqrt{H^2SAT})$ and $O(H^2SA \log T)$, respectively. Recently, Qiao et al. [2022] showed that to achieve the optimal $\widetilde{O}(\sqrt{T})$ regret, both the global switching cost upper and lower bounds are of order $HSA \log \log T$. Under linear MDP, Gao et al. [2021] applied doubling trick to LSVI-UCB and arrived at regret bound $\widetilde{O}(\sqrt{d^3H^3T})$ while global switching cost is $O(dH \log T)$. This result is generalized by Wang et al. [2021] to work for

arbitrary switching cost budget. Huang et al. [2022] managed to do pure exploration under linear MDP within $O(dH)$ switches.

**Batched bandits and RL.** In batched bandits problems, the agent decides a sequence of arms and observes the reward of each arm after all arms in that sequence are pulled. More formally, at the beginning of each batch, the agent decides a list of arms to be pulled. Afterwards, a list of (arm,reward) pairs is given to the agent. Then the agent decides about the next batch [Esfandiari et al., 2021]. The batch sizes could be chosen non-adaptively or adaptively. In a non-adaptive algorithm, the batch sizes should be decided before the algorithm starts, while in an adaptive algorithm, the batch sizes may depend on the previous observations. Under multi-armed bandits with $A$ arms and $T$ episodes, Cesa-Bianchi et al. [2013] designed an algorithm with $\widetilde{O}(\sqrt{AT})$ regret using $O(\log \log T)$ batches. Perchet et al. [2016] proved a regret lower bound of $\Omega(T^{\frac{1}{1-2^{1-M}}})$ for algorithms within $M$ batches under 2-armed bandits setting, which means $\Omega(\log \log T)$ batches are necessary for a regret bound of $\widetilde{O}(\sqrt{T})$. The result is generalized to $K$-armed bandits by Gao et al. [2019]. Under stochastic linear bandits, Han et al. [2020] designed an algorithm that has regret bound $\widetilde{O}(\sqrt{T})$ while running in $O(\log \log T)$ batches. Ruan et al. [2021] improved this result by using weaker assumptions. For batched RL setting, Qiao et al. [2022] showed that their algorithm uses the optimal $O(H + \log \log T)$ batches to achieve the optimal $\widetilde{O}(\sqrt{T})$ regret. Recently, the regret bound and computational efficiency is improved by Zhang et al. [2022] through incorporating the idea of optimal experimental design. The deployment efficient algorithms for pure exploration by Huang et al. [2022] also satisfy the definition of batched RL.

# B  Generalization of G-optimal design

**Traditional G-optimal design.** We first briefly introduce the problem setup of G-optimal design. Assume there is some (possibly infinite) set $\mathcal{A} \subseteq \mathbb{R}^d$, let $\pi : \mathcal{A} \to [0,1]$ be a distribution on $\mathcal{A}$ so that $\sum_{a \in \mathcal{A}} \pi(a) = 1$. $V(\pi) \in \mathbb{R}^{d \times d}$ and $g(\pi) \in \mathbb{R}$ are given by

$$V(\pi) = \sum_{a \in \mathcal{A}} \pi(a) a a^\top, \quad g(\pi) = \max_{a \in \mathcal{A}} \|a\|^2_{V(\pi)^{-1}}.$$

The problem of finding a design $\pi$ that minimises $g(\pi)$ is called the **G-optimal design** problem. G-optimal design has wide application in regression problems and it can solve the linear bandit problem [Lattimore and Szepesvári, 2020]. However, traditional G-optimal design can not tackle our problem under linear MDP where we can only choose $\pi$ instead of choosing the feature vector $\phi$ directly.

In this section, we generalize the well-known G-optimal design for our purpose under linear MDP. Consider the following problem: Under some fixed linear MDP, given a fixed finite policy set $\Pi$, we want to select a policy $\pi_0$ from $\Delta(\Pi)$ (distribution over policy set $\Pi$) to minimize the following term:

$$\max_{\pi \in \Pi} \mathbb{E}_\pi \phi(s_h, a_h)^\top (\mathbb{E}_{\pi_0} \phi_h \phi_h^\top)^{-1} \phi(s_h, a_h), \tag{2}$$

where the $s_h, a_h$ follows the distribution according to $\pi$ and the $\phi_h$ follows the distribution of policy $\pi_0$. We first consider its two special cases.

**Special case 1.** If the MDP is deterministic, then given any fixed deterministic policy $\pi$, the trajectory generated from this $\pi$ is deterministic. Therefore the feature $\phi_h$ at layer $h$ is also deterministic. We denote the feature at layer $h$ from running policy $\pi$ by $\phi_{\pi,h}$. In this case, the previous problem (2) reduces to

$$\min_{\pi_0 \in \Delta(\Pi)} \max_{\pi \in \Pi} \phi_{\pi,h}^\top (\mathbb{E}_{\pi_0} \phi_h \phi_h^\top)^{-1} \phi_{\pi,h}, \tag{3}$$

which can be characterized by the traditional G-optimal design, for more details please refer to Kiefer and Wolfowitz [1960] and chapter 21 of Lattimore and Szepesvári [2020]. According to Theorem 21.1 of Lattimore and Szepesvári [2020], the minimization of (3) can be bounded by $d$, which is the dimension of the feature map $\phi$.

**Special case 2.** When the linear MDP is actually a tabular MDP with finite state set $|\mathcal{S}| = S$ and finite action set $|\mathcal{A}| = A$, the feature map reduces to canonical basis in $\mathbb{R}^d = \mathbb{R}^{SA}$ with $\phi(s,a) = e_{(s,a)}$

[Jin et al., 2020b]. Let $d_h^\pi(s, a) = \mathbb{P}_\pi(s_h = s, a_h = a)$ denote the occupancy measure, then the previous optimization problem (2) reduces to

$$\min_{\pi_0 \in \Delta(\Pi)} \max_{\pi \in \Pi} \sum_{(s,a) \in \mathcal{S} \times \mathcal{A}} \frac{d_h^\pi(s, a)}{d_h^{\pi_0}(s, a)}. \tag{4}$$

Such minimization problem corresponds to finding a policy $\pi_0$ that can cover all policies from the policy set $\Pi$. According to Lemma 1 in Zhang et al. [2022] (we only use the case where $m = 1$), the minimization of (4) can be bounded by $d = SA$.

Different from these two special cases, under our problem setup (general linear MDP), the feature map can be much more complex than canonical basis and running each $\pi$ will lead to a distribution over the feature map space rather than a fixed single feature. Next, we formalize the problem setup and present the theorem. We are given finite policy set $\Pi$ and finite action set $\Phi$ (we only consider finite action set, the general case can be proven similarly by passing to the limit [Lattimore and Szepesvári, 2020]), where each $\pi \in \Pi$ is a distribution over $\Phi$ (with $\pi(a)$ denoting the probability of choosing action $a$) and each action $a \in \Phi$ is a vector in $\mathbb{R}^d$. In addition, $\mu$ can be any distribution over $\Pi$. In the following part, we characterize $\mu$ as a vector in $\mathbb{R}^{|\Pi|}$ with $\mu(\pi)$ denoting the probability of choosing policy $\pi$. Let $\Lambda(\pi) = \sum_{a \in \Phi} \pi(a) a a^\top$ and $V(\mu) = \sum_{\pi \in \Pi} \mu(\pi) \Lambda(\pi) = \sum_{\pi \in \Pi} \mu(\pi) \sum_{a \in \Phi} \pi(a) a a^\top$. The function we want to minimize is $g(\mu) = \max_{\pi \in \Pi} \sum_{a \in \Phi} \pi(a) a^\top V(\mu)^{-1} a$.

**Theorem B.1.** *Define the set* $\widehat{\Phi} = \{a \in \Phi : \exists \pi \in \Pi, \pi(a) > 0\}$. *If* $span(\widehat{\Phi}) = \mathbb{R}^d$, *there exists a distribution* $\mu^\star$ *over* $\Pi$ *such that* $g(\mu^\star) \le d$.

*Proof of Theorem B.1.* Define $f(\mu) = \log \det V(\mu)$ and take $\mu^\star$ to be

$$\mu^\star = \arg \max_\mu f(\mu).$$

According to Exercise 21.2 of Lattimore and Szepesvári [2020], $f$ is concave. Besides, according to Exercise 21.1 of Lattimore and Szepesvári [2020], we have

$$\frac{d}{dt} \log \det(A(t)) = \frac{1}{\det(A(t))} Tr(adj(A) \frac{d}{dt} A(t)) = Tr(A^{-1} \frac{d}{dt} A(t)).$$

Plugging $f$ in, we directly have:

$$(\nabla f(\mu))_\pi = Tr(V(\mu)^{-1} \Lambda(\pi)) = \sum_{a \in \Phi} \pi(a) a^\top V(\mu)^{-1} a.$$

In addition, by direct calculation, for any feasible $\mu$,

$$\sum_{\pi \in \Pi} \mu(\pi)(\nabla f(\mu))_\pi = Tr(\sum_{\pi \in \Pi} \mu(\pi) \sum_{a \in \Phi} \pi(a) a a^\top V(\mu)^{-1}) = Tr(I_d) = d.$$

Since $\mu^\star$ is the maximizer of $f$, by first order optimality criterion, for any feasible $\mu$,

$$\begin{aligned}
0 \ge &\langle \nabla f(\mu^\star), \mu - \mu^\star \rangle \\
= &\sum_{\pi \in \Pi} \mu(\pi) \sum_{a \in \Phi} \pi(a) a^\top V(\mu^\star)^{-1} a - \sum_{\pi \in \Pi} \mu^\star(\pi) \sum_{a \in \Phi} \pi(a) a^\top V(\mu^\star)^{-1} a \\
= &\sum_{\pi \in \Pi} \mu(\pi) \sum_{a \in \Phi} \pi(a) a^\top V(\mu^\star)^{-1} a - d.
\end{aligned}$$

For any $\pi \in \Pi$, we can choose $\mu$ to be Dirac at $\pi$, which proves that for any $\pi \in \Pi$, $\sum_{a \in \Phi} \pi(a) a^\top V(\mu^\star)^{-1} a \le d$. Due to the definition of $g(\mu^\star)$, we have $g(\mu^\star) \le d$. $\qquad \square$

**Remark B.2.** *By replacing the action set* $\Phi$ *with the set of all feasible features at layer* $h$, *Theorem B.1 shows that for any linear MDP and fixed policy set* $\Pi$,

$$\min_{\pi_0 \in \Delta(\Pi)} \max_{\pi \in \Pi} \mathbb{E}_\pi \phi(s_h, a_h)^\top (\mathbb{E}_{\pi_0} \phi_h \phi_h^\top)^{-1} \phi(s_h, a_h) \le d. \tag{5}$$

*This theorem serves as one of the critical theoretical bases for our analysis.*

**Remark B.3.** *Although the proof is similar to Theorem 21.1 of Lattimore and Szepesvári [2020], our Theorem B.1 is more general since it also holds under the case where each $\pi$ will generate a distribution over the action space. In contrast, G-optimal design is a special case of our setting where each $\pi$ will generate a fixed action from the action space.*

Knowing the existence of such covering policy, the next lemma provides some properties of the solution of (2) under some additional assumption.

**Lemma B.4.** *Let $\pi^\star = \arg\min_{\pi_0 \in \Delta(\Pi)} \max_{\pi \in \Pi} \mathbb{E}_\pi \phi(s_h, a_h)^\top (\mathbb{E}_{\pi_0} \phi_h \phi_h^\top)^{-1} \phi(s_h, a_h)$. Assume that $\sup_{\pi \in \Delta(\Pi)} \lambda_{\min}(\mathbb{E}_\pi \phi_h \phi_h^\top) \geq \lambda^\star$, then it holds that*

$$\lambda_{\min}(\mathbb{E}_{\pi^\star} \phi_h \phi_h^\top) \geq \frac{\lambda^\star}{d}, \tag{6}$$

*where $d$ is the dimension of $\phi$ and $\lambda_{\min}$ denotes the minimum eigenvalue.*

Before we state the proof, we provide the description of the special case where the MDP is a tabular MDP. The condition implies that there exists some policy $\widetilde{\pi} \in \Delta(\Pi)$ such that for any $s, a \in \mathcal{S} \times \mathcal{A}$, $d_h^{\widetilde{\pi}}(s, a) \geq \lambda^\star$, where $d_h^\pi(\cdot, \cdot)$ is occupancy measure. Due to Theorem B.1, $\pi^\star$ satisfies that

$$\max_{\pi \in \Pi} \sum_{(s,a) \in \mathcal{S} \times \mathcal{A}} \frac{d_h^\pi(s, a)}{d_h^{\pi^\star}(s, a)} \leq SA.$$

For any $(s, a) \in \mathcal{S} \times \mathcal{A}$, choose $\pi_{s,a} = \arg\max_{\pi \in \Pi} d_h^\pi(s, a)$ and $d_h^{\pi_{s,a}}(s, a) \geq d_h^{\widetilde{\pi}}(s, a) \geq \lambda^\star$. Therefore, it holds that $d_h^{\pi^\star}(s, a) \geq \frac{\lambda^\star}{SA}$ for any $s, a$, which is equivalent to the conclusion of (6).

*Proof of Lemma B.4.* If the conclusion (6) does not hold, we have $\lambda_{\min}(\mathbb{E}_{\pi^\star} \phi_h \phi_h^\top) < \frac{\lambda^\star}{d}$, which implies that $\lambda_{\max}((\mathbb{E}_{\pi^\star} \phi_h \phi_h^\top)^{-1}) > \frac{d}{\lambda^\star}$. Denote the eigenvalues of $(\mathbb{E}_{\pi^\star} \phi_h \phi_h^\top)^{-1}$ by $0 < \lambda_1 \leq \lambda_2 \leq \cdots \leq \lambda_d$. There exists a set of orthogonal and normalized vectors $\{\bar{\phi}_i\}_{i \in [d]}$ such that $\bar{\phi}_i$ is a corresponding eigenvector of $\lambda_i$.

According to the condition, there exists $\widetilde{\pi} \in \Delta(\Pi)$ such that $\lambda_{\min}(\mathbb{E}_{\widetilde{\pi}} \phi_h \phi_h^\top) \geq \lambda^\star$. Therefore, for any $\phi \in \mathbb{R}^d$ with $\|\phi\|_2 = 1$, $\phi^\top (\mathbb{E}_{\widetilde{\pi}} \phi_h \phi_h^\top) \phi = \mathbb{E}_{\widetilde{\pi}} (\phi_h^\top \phi)^2 \geq \lambda^\star$. Now we consider $\mathbb{E}_{\widetilde{\pi}} \phi_h^\top (\mathbb{E}_{\pi^\star})^{-1} \phi_h$, where $\mathbb{E}_{\pi^\star}$ is short for $\mathbb{E}_{\pi^\star} \phi_h \phi_h^\top$. It holds that:

$$
\begin{aligned}
\mathbb{E}_{\widetilde{\pi}} \phi_h^\top (\mathbb{E}_{\pi^\star})^{-1} \phi_h &= \mathbb{E}_{\widetilde{\pi}} [\sum_{i=1}^d (\phi_h^\top \bar{\phi}_i) \bar{\phi}_i]^\top (\mathbb{E}_{\pi^\star})^{-1} [\sum_{i=1}^d (\phi_h^\top \bar{\phi}_i) \bar{\phi}_i] \\
&= \mathbb{E}_{\widetilde{\pi}} \sum_{i=1}^d (\phi_h^\top \bar{\phi}_i)^2 \bar{\phi}_i^\top (\mathbb{E}_{\pi^\star})^{-1} \bar{\phi}_i \\
&\geq \mathbb{E}_{\widetilde{\pi}} (\phi_h^\top \bar{\phi}_d)^2 \bar{\phi}_d^\top (\mathbb{E}_{\pi^\star})^{-1} \bar{\phi}_d \\
&> \lambda^\star \times \frac{d}{\lambda^\star} = d,
\end{aligned}
$$

where the first equation is due to the fact that $\{\bar{\phi}_i\}_{i \in [d]}$ forms a set of normalized basis. The second equation results from the definition of eigenvectors. The last inequality is because our assumption $(\lambda_{\max}((\mathbb{E}_{\pi^\star} \phi_h \phi_h^\top)^{-1}) > \frac{d}{\lambda^\star})$ and condition $(\forall \|\phi\|_2 = 1, \phi^\top (\mathbb{E}_{\widetilde{\pi}} \phi_h \phi_h^\top) \phi = \mathbb{E}_{\widetilde{\pi}} (\phi_h^\top \phi)^2 \geq \lambda^\star)$.

Finally, since this leads to contradiction with Theorem B.1, the proof is complete. $\qquad\square$

## C  Construction of policy sets

In this section, we construct policy sets given the feature map $\phi(\cdot, \cdot)$. We begin with several technical lemmas.

## C.1 Technical lemmas

**Lemma C.1** (Covering Number of Euclidean Ball [Jin et al., 2020b]). *For any $\epsilon > 0$, the $\epsilon$-covering number of the Euclidean ball in $\mathbb{R}^d$ with radius $R > 0$ is upper bounded by $(1 + \frac{2R}{\epsilon})^d$.*

**Lemma C.2** (Lemma B.1 of Jin et al. [2020b]). *Let $w_h^\pi$ denote the set of weights such that $Q_h^\pi(s,a) = \langle \phi(s,a), w_h^\pi \rangle$. Then $\|w_h^\pi\|_2 \leq 2H\sqrt{d}$.*

**Lemma C.3** (Advantage Decomposition). *For any MDP with fixed initial state $s_1$, for any policy $\pi$, it holds that*

$$V_1^\star(s_1) - V_1^\pi(s_1) = \mathbb{E}_\pi \sum_{h=1}^H [V_h^\star(s_h) - Q_h^\star(s_h, a_h)],$$

*here the expectation means that $s_h, a_h$ follows the distribution generated by $\pi$.*

*Proof of Lemma C.3.*

$$\begin{aligned}
V_1^\star(s_1) - V_1^\pi(s_1) =& \mathbb{E}_\pi[V_1^\star(s_1) - Q_1^\star(s_1, a_1)] + \mathbb{E}_\pi[Q_1^\star(s_1, a_1) - Q_1^\pi(s_1, a_1)] \\
=& \mathbb{E}_\pi[V_1^\star(s_1) - Q_1^\star(s_1, a_1)] + \mathbb{E}_{s_1, a_1 \sim \pi}\Big[\sum_{s' \in \mathcal{S}} P_1(s'|s_1, a_1)(V_2^\star(s') - V_2^\pi(s'))\Big] \\
=& \mathbb{E}_\pi[V_1^\star(s_1) - Q_1^\star(s_1, a_1)] + \mathbb{E}_{s_2 \sim \pi}[V_2^\star(s_2) - V_2^\pi(s_2)] \\
=& \cdots \\
=& \mathbb{E}_\pi \sum_{h=1}^H [V_h^\star(s_h) - Q_h^\star(s_h, a_h)],
\end{aligned}$$

where the second equation is because of Bellman Equation and the forth equation results from applying the decomposition recursively from $h = 1$ to $H$. □

**Lemma C.4** (Elliptical Potential Lemma, Lemma 26 of Agarwal et al. [2020]). *Consider a sequence of $d \times d$ positive semi-definite matrices $X_1, \cdots, X_T$ with $\max_t Tr(X_t) \leq 1$ and define $M_0 = I, \cdots, M_t = M_{t-1} + X_t$. Then*

$$\sum_{t=1}^T Tr(X_t M_{t-1}^{-1}) \leq 2d \log(1 + \frac{T}{d}).$$

## C.2 Construction of policies to evaluate

We construct the policy set $\Pi^{eval}$ given feature map $\phi(\cdot, \cdot)$. The policy set $\Pi^{eval}$ satisfies that for any feasible linear MDP with feature map $\phi$, $\Pi^{eval}$ contains one near-optimal policy of this linear MDP. We begin with the construction.

**Construction of $\Pi^{eval}$.** Given $\epsilon > 0$, let $\mathcal{W}$ be a $\frac{\epsilon}{2H}$-cover of the Euclidean ball $\mathcal{B}^d(2H\sqrt{d}) := \{x \in \mathbb{R}^d : \|x\|_2 \leq 2H\sqrt{d}\}$. Next, we construct the Q-function set $\mathcal{Q} = \{\bar{Q}(s,a) = \phi(s,a)^\top w : w \in \mathcal{W}\}$. Then the policy set at layer $h$ is defined as $\forall\, h \in [H]$, $\Pi_h = \{\pi(s) = \arg\max_{a \in \mathcal{A}} \bar{Q}(s,a) | \bar{Q} \in \mathcal{Q}\}$, with ties broken arbitrarily. Finally, the policy set $\Pi_\epsilon^{eval}$ is $\Pi_\epsilon^{eval} = \Pi_1 \times \Pi_2 \times \cdots \times \Pi_H$.

**Lemma C.5.** *The policy set $\Pi_\epsilon^{eval}$ satisfies that for any $h \in [H]$,*

$$\log |\Pi_h| \leq d \log(1 + \frac{8H^2\sqrt{d}}{\epsilon}) = \widetilde{O}(d). \tag{7}$$

*In addition, for any linear MDP with feature map $\phi(\cdot, \cdot)$, there exists $\pi = (\pi_1, \pi_2, \cdots, \pi_H)$ such that $\pi_h \in \Pi_h$ for all $h \in [H]$ and $V^\pi \geq V^\star - \epsilon$.*

*Proof of Lemma C.5.* Since $\mathcal{W}$ is a $\frac{\epsilon}{2H}$-covering of Euclidean ball, by Lemma C.1 we have

$$\log |\mathcal{W}| \leq d \log(1 + \frac{8H^2\sqrt{d}}{\epsilon}).$$

In addition, for any $w$ in $\mathcal{W}$, there is at most one corresponding $Q \in \mathcal{Q}$ and one $\pi_h \in \Pi_h$. Therefore, it holds that for any $h \in [H]$,

$$\log |\Pi_h| \leq \log |\mathcal{Q}| \leq \log |\mathcal{W}| \leq d \log(1 + \frac{8H^2\sqrt{d}}{\epsilon}).$$

For any linear MDP, according to Lemma C.2, the optimal Q-function can be written as:

$$Q_h^\star(s,a) = \langle \phi(s,a), w_h^\star \rangle,$$

with $\|w_h^\star\|_2 \leq 2H\sqrt{d}$. Since $\mathcal{W}$ is $\frac{\epsilon}{2H}$-covering of the Euclidean ball, for any $h \in [H]$ there exists $\bar{w}_h \in \mathcal{W}$ such that $\|\bar{w}_h - w_h^\star\|_2 \leq \frac{\epsilon}{2H}$. Select $\bar{Q}_h(s,a) = \phi(s,a)^\top \bar{w}_h$ from $\mathcal{Q}$ and $\pi_h(s) = \arg\max_{a \in \mathcal{A}} \bar{Q}_h(s,a)$ from $\Pi_h$. Note that for any $h,s,a \in [H] \times \mathcal{S} \times \mathcal{A}$,

$$|Q_h^\star(s,a) - \bar{Q}_h(s,a)| \leq \|\phi(s,a)\|_2 \cdot \|w_h^\star - \bar{w}_h\|_2 \leq \frac{\epsilon}{2H}. \tag{8}$$

Let $\pi = (\pi_1, \pi_2, \cdots, \pi_H)$, now we prove that this $\pi$ is $\epsilon$-optimal.

Denote the optimal policy under this linear MDP by $\pi^\star$, then we have for any $s,h \in \mathcal{S} \times [H]$,

$$Q_h^\star(s, \pi_h^\star(s)) - Q_h^\star(s, \pi_h(s))$$
$$= [Q_h^\star(s, \pi_h^\star(s)) - \bar{Q}_h(s, \pi_h^\star(s))] + [\bar{Q}_h(s, \pi_h^\star(s)) - \bar{Q}_h(s, \pi_h(s))] + [\bar{Q}_h(s, \pi_h(s)) - Q_h^\star(s, \pi_h(s))]$$
$$\leq \frac{\epsilon}{2H} + 0 + \frac{\epsilon}{2H} = \frac{\epsilon}{H},$$

$$\tag{9}$$

where the inequality results from the definition of $\pi_h$ and (8).

Now we apply the advantage decomposition (Lemma C.3), it holds that:

$$V_1^\star(s_1) - V_1^\pi(s_1) = \mathbb{E}_\pi \sum_{h=1}^{H} [V_h^\star(s_h) - Q_h^\star(s_h, a_h)]$$
$$\leq H \cdot \frac{\epsilon}{H} = \epsilon,$$

where the inequality comes from (9). $\qquad \square$

**Remark C.6.** *Our concurrent work Wagenmaker and Jamieson [2022] also applies the idea of policy discretization. However, to cover $\epsilon$-optimal policies of all linear MDPs, the size of their policy set is $\log|\Pi_\epsilon| \leq \widetilde{O}(dH^2 \cdot \log\frac{1}{\epsilon})$ (stated in Corollary 1 of Wagenmaker and Jamieson [2022]). In comparison, our $\Pi_\epsilon^{eval}$ satisfies that $\log|\Pi_\epsilon^{eval}| \leq H\log|\Pi_1| \leq \widetilde{O}(dH \cdot \log\frac{1}{\epsilon})$, which improves their results by a factor of $H$. Such improvement is done by applying advantage decomposition. Finally, by plugging in our $\Pi_\epsilon^{eval}$ into Corollary 2 of Wagenmaker and Jamieson [2022], we can directly improve their worst-case bound by a factor of $H$.*

### C.3 Construction of explorative policies

Given the feature map $\phi(\cdot, \cdot)$ and the condition that for any $h \in [H]$, $\sup_\pi \lambda_{\min}(\mathbb{E}_\pi \phi_h \phi_h^\top) \geq \lambda^\star$ where $\pi$ can be any policy, we construct a finite policy set $\Pi^{exp}$ that covers explorative policies under any feasible linear MDPs. Such exploratory is formalized as for any linear MDP and $h \in [H]$, there exists some policy $\pi$ in $\Delta(\Pi^{exp})$ such that $\lambda_{\min}(\mathbb{E}_\pi \phi_h \phi_h^\top)$ is large enough. We begin with the construction.

**Construction of $\Pi^{exp}$.** Given $\epsilon > 0$, consider all reward functions that can be represented as

$$r(s,a) = \sqrt{\phi(s,a)^\top (I + \Sigma)^{-1} \phi(s,a)}, \tag{10}$$

where $\Sigma$ is positive semi-definite. According to Lemma D.6 of Jin et al. [2020b], we can construct a $\frac{\epsilon}{2H}$-cover $\mathcal{R}_\epsilon$ of all such reward functions while the size of $\mathcal{R}_\epsilon$ satisfies $\log|\mathcal{R}_\epsilon| \leq d^2 \log(1 + \frac{32H^2\sqrt{d}}{\epsilon^2})$.

For all $h \in [H]$, denote $\Pi_{h,\epsilon}^1 = \{\pi(s) = \arg\max_{a \in \mathcal{A}} r(s,a) | r \in \mathcal{R}_\epsilon\}$ with ties broken arbitrarily. Meanwhile, denote the policy set $\Pi_h$ (w.r.t $\epsilon$) in the previous Section C.2 by $\Pi_{h,\epsilon}^2$. Finally, let $\Pi_{h,\epsilon} = \Pi_{h,\epsilon}^1 \cup \Pi_{h,\epsilon}^2$ be the policy set for layer $h$. The whole policy set is the product of these $h$ policy sets, $\Pi_\epsilon^{exp} = \Pi_{1,\epsilon} \times \cdots \times \Pi_{H,\epsilon}$.

**Lemma C.7.** *For any $\epsilon > 0$, we have $\Pi_\epsilon^{eval} \subseteq \Pi_\epsilon^{exp}$. In addition, $\log |\Pi_{h,\epsilon}| \leq 2d^2 \log(1 + \frac{32H^2\sqrt{d}}{\epsilon^2})$. For any reward $r$ that is the form of (10) and $h \in [H]$, there exists a policy $\bar{\pi} \in \Pi_\epsilon^{exp}$ such that*

$$\mathbb{E}_{\bar{\pi}} r(s_h, a_h) \geq \sup_\pi \mathbb{E}_\pi r(s_h, a_h) - \epsilon.$$

*Proof of Lemma C.7.* The conclusion that $\Pi_\epsilon^{eval} \subseteq \Pi_\epsilon^{exp}$ is because of our construction: $\Pi_{h,\epsilon} = \Pi_{h,\epsilon}^1 \cup \Pi_{h,\epsilon}^2$.

In addition,

$$\log |\Pi_{h,\epsilon}| \leq \log |\Pi_{h,\epsilon}^1| + \log |\Pi_{h,\epsilon}^2| \leq \log |\mathcal{R}_\epsilon| + d\log(1 + \frac{8H^2\sqrt{d}}{\epsilon}) \leq 2d^2 \log(1 + \frac{32H^2\sqrt{d}}{\epsilon^2}).$$

Consider the optimal Q-function under reward function $r(s_h, a_h)$ (reward is always 0 at other layers). We have $Q_h^\star(s, a) = r(s, a)$ and for $i \leq h - 1$,

$$\begin{aligned}
Q_i^\star(s, a) &= 0 + \sum_{s' \in \mathcal{S}} \langle \phi(s, a), \mu_i(s') \rangle V_{i+1}^\star(s') \\
&= \langle \phi(s, a), \sum_{s' \in \mathcal{S}} \mu_i(s') V_{i+1}^\star(s') \rangle \\
&= \langle \phi(s, a), w_i^\star \rangle,
\end{aligned}$$

for some $w_i^\star \in \mathbb{R}^d$ with $\|w_i^\star\|_2 \leq 2\sqrt{d}$. The first equation is because of Bellman Equation and our design of reward function.

Since $Q_h^\star$ is covered by $\mathcal{R}_\epsilon$ up to $\frac{\epsilon}{2H}$ accuracy while $Q_i^\star$ ($i \leq h - 1$) is covered by $\mathcal{Q}$ in section C.2 up to $\frac{\epsilon}{2H}$ accuracy, with identical proof to Lemma C.5, the last conclusion holds. $\square$

**Lemma C.8.** *Assume $\sup_\pi \lambda_{\min}(\mathbb{E}_\pi \phi_h \phi_h^\top) \geq \lambda^\star$, if $\epsilon \leq \frac{\lambda^\star}{4}$, we have*

$$\sup_{\pi \in \Delta(\Pi_\epsilon^{exp})} \lambda_{\min}(\mathbb{E}_\pi \phi_h \phi_h^\top) \geq \frac{(\lambda^\star)^2}{64d \log(1/\lambda^\star)}.$$

*Proof of Lemma C.8.* Fix $t = \frac{64d \log(1/\lambda^\star)}{(\lambda^\star)^2}$, we construct the following policies:
$\pi_1$ is arbitrary policy in $\Pi_\epsilon^{exp}$.
For any $i \in [t]$, $\Sigma_i = \sum_{j=1}^i \mathbb{E}_{\pi_j} \phi_h \phi_h^\top$, $r_i(s, a) = \sqrt{\phi(s, a)^\top (I + \Sigma_i)^{-1} \phi(s, a)}$. Due to Lemma C.7, there exists policy $\pi_{i+1} \in \Pi_\epsilon^{exp}$ such that $\mathbb{E}_{\pi_{i+1}} r_i(s_h, a_h) \geq \sup_\pi \mathbb{E}_\pi r_i(s_h, a_h) - \epsilon$.

The following inequality holds:

$$\begin{aligned}
&\sum_{i=1}^t \mathbb{E}_{\pi_i} \sqrt{\phi_h^\top (I + \Sigma_{i-1})^{-1} \phi_h} \\
&\leq \sum_{i=1}^t \sqrt{\mathbb{E}_{\pi_i} \phi_h^\top (I + \Sigma_{i-1})^{-1} \phi_h} \\
&\leq \sqrt{t \cdot \sum_{i=1}^t \mathbb{E}_{\pi_i} \phi_h^\top (I + \Sigma_{i-1})^{-1} \phi_h} \\
&\leq \sqrt{t \cdot \sum_{i=1}^t Tr(\mathbb{E}_{\pi_i} \phi_h \phi_h^\top (I + \Sigma_{i-1})^{-1})} \\
&\leq \sqrt{2dt \log(1 + \frac{t}{d})},
\end{aligned} \tag{11}$$

where the second inequality holds because of Cauchy-Schwarz inequality and the last inequality holds due to Lemma C.4.

Therefore, we have that $\sup_\pi \mathbb{E}_\pi \sqrt{\phi_h^\top (I + \Sigma_{t-1})^{-1} \phi_h} \le \sqrt{\frac{2d \log(1+t/d)}{t}} + \epsilon \le \frac{\lambda^\star}{2}$ because of our choice of $\epsilon \le \frac{\lambda^\star}{4}$ and $t = \frac{64d \log(1/\lambda^\star)}{(\lambda^\star)^2}$. According to Lemma E.14[12] of Huang et al. [2022], we have that $\lambda_{\min}(\Sigma_{t-1}) \ge 1$.

Finally, choose $\pi = unif(\{\pi_i\}_{i \in [t-1]})$, we have $\pi \in \Delta(\Pi_\epsilon^{exp})$ and

$$\lambda_{\min}(\mathbb{E}_\pi \phi_h \phi_h^\top) \ge \frac{(\lambda^\star)^2}{64d \log(1/\lambda^\star)}.$$

$\square$

## C.4 A summary

| Policy sets | Cardinality | Description | Relationship with each other |
|---|---|---|---|
| The set of all policies | Infinity | The largest possible policy set | Contains the following two sets |
| Explorative policies: $\Pi_\epsilon^{exp}$ | $\log|\Pi_{\epsilon,h}^{exp}| = \widetilde{O}(d^2)$ | Sufficient for exploration | Subset of all policies |
| Policies to evaluate: $\Pi_\epsilon^{eval}$ | $\log|\Pi_{\epsilon,h}^{eval}| = \widetilde{O}(d)$ | Uniform policy evaluation over $\Pi_\epsilon^{eval}$ is sufficient for policy identification | Subset of $\Pi_\epsilon^{exp}$ |

Table 2: Comparison of different policy sets.

We compare the relationship between different policy sets in the Table 2 above. In summary, given the feature map $\phi(\cdot, \cdot)$ of linear MDP and any accuracy $\epsilon$, we can construct policy set $\Pi^{eval}$ which satisfies that $\log|\Pi_h^{eval}| = \widetilde{O}(d)$. At the same time, for any linear MDP, the policy set $\Pi^{eval}$ is guaranteed to contain one near-optimal policy. Therefore, it suffices to estimate the value functions of all policies in $\Pi^{eval}$ accurately.

Similarly, given the feature map $\phi(\cdot, \cdot)$ and some $\epsilon$ that is small enough compared to $\lambda^\star$, we can construct policy set $\Pi^{exp}$ which satisfies that $\log|\Pi_h^{exp}| = \widetilde{O}(d^2)$. At the same time, for any linear MDP, the policy set $\Pi^{exp}$ is guaranteed to contain explorative policies for all layers, which means that it suffices to do exploration using only policies from $\Pi^{exp}$.

# D Estimation of value functions

According to the construction of $\Pi^{eval}$ in Section C.2 and Lemma C.5, it suffices to estimate the value functions of policies in $\Pi^{eval}$. In this section, we design an algorithm to estimate the value functions of any policy in $\Pi^{eval}$ given any reward function. Recall that for accuracy $\epsilon_0$, we denote the policy set constructed in Section C.2 by $\Pi_{\epsilon_0}^{eval}$ and the policy set for layer $h$ is denoted by $\Pi_{\epsilon_0,h}^{eval}$.

## D.1 The algorithm

---

**Algorithm 3** Estimation of $V^\pi(r)$ given exploration data (EstimateV)

---

1: **Input:** Policy to evaluate $\pi \in \Pi_{\epsilon_0}^{eval}$. Linear reward function $r = \{r_h\}_{h \in [H]}$ bounded in $[0, 1]$. Exploration data $\{s_h^n, a_h^n\}_{(h,n) \in [H] \times [N]}$. Initial state $s_1$.
2: **Initialization:** $Q_{H+1}(\cdot, \cdot) \leftarrow 0$, $V_{H+1}(\cdot) \leftarrow 0$.
3: **for** $h = H, H-1, \ldots, 1$ **do**
4: $\quad \Lambda_h \leftarrow I + \sum_{n=1}^N \phi(s_h^n, a_h^n) \phi(s_h^n, a_h^n)^\top$.
5: $\quad \bar{w}_h \leftarrow (\Lambda_h)^{-1} \sum_{n=1}^N \phi(s_h^n, a_h^n) V_{h+1}(s_{h+1}^n)$.
6: $\quad Q_h(\cdot, \cdot) \leftarrow (\phi(\cdot, \cdot)^\top \bar{w}_h + r_h(\cdot, \cdot))_{[0,H]}$.
7: $\quad V_h(\cdot) \leftarrow Q_h(\cdot, \pi_h(\cdot))$.
8: **end for**
9: **Output:** $V_1(s_1)$.

---

[12]Our condition that $\sup_\pi \lambda_{\min}(\mathbb{E}_\pi \phi_h \phi_h^\top) \ge \lambda^\star$ implies that for any $u \in \mathbb{R}^d$ with $\|u\|_2 = 1$, $\max_\pi \mathbb{E}_\pi (\phi_h^\top u)^2 \ge \lambda^\star$. Therefore, the proof of Lemma E.14 of Huang et al. [2022] holds by plugging in $c = 1$.

Algorithm 3 takes policy $\pi$ from $\Pi_{\epsilon_0}^{eval}$ and linear reward function $r$ as input, and uses LSVI to estimate the value function of this given policy and given reward function. From layer $H$ to layer 1, we calculate $\Lambda_h$ and $\bar{w}_h$ to estimate $Q_h^\pi$ in line 6. In addition, according to our construction in Section C.2, all policies in $\Pi_{\epsilon_0}^{eval}$ are deterministic, which means we can use line 7 to approximate $V_h^\pi$. Algorithm 3 looks similar to Algorithm 2 in Wang et al. [2020]. However, there are two key differences. First, Algorithm 2 of Wang et al. [2020] aims to find near optimal policy for each reward function while we do policy evaluation for each reward and policy. In addition, different from their approach, we do not use optimism, which means we do not need to cover the bonus term. This is the main reason why we can save a factor of $\sqrt{d}$.

## D.2 Technical lemmas

**Lemma D.1** (Lemma D.4 of Jin et al. [2020b]). *Let $\{x_\tau\}_{\tau=1}^\infty$ be a stochastic process on state space $\mathcal{S}$ with corresponding filtration $\{\mathcal{F}_\tau\}_{\tau=0}^\infty$. Let $\{\phi_\tau\}_{\tau=1}^\infty$ be an $\mathbb{R}^d$-valued stochastic process where $\phi_\tau \in \mathcal{F}_{\tau-1}$, and $\|\phi_\tau\| \leq 1$. Let $\Lambda_k = I + \sum_{\tau=1}^k \phi_\tau \phi_\tau^\top$. Then for any $\delta > 0$, with probability at least $1 - \delta$, for all $k \geq 0$, and any $V \in \mathcal{V}$ so that $\sup_x |V(x)| \leq H$, we have:*

$$\left\| \sum_{\tau=1}^k \phi_\tau \{V(x_\tau) - \mathbb{E}[V(x_\tau)|\mathcal{F}_{\tau-1}]\} \right\|_{\Lambda_k^{-1}}^2 \leq 4H^2 \left[ \frac{d}{2} \log(k+1) + \log(\frac{\mathcal{N}_\epsilon}{\delta}) \right] + 8k^2\epsilon^2,$$

*where $\mathcal{N}_\epsilon$ is the $\epsilon$-covering number of $\mathcal{V}$ with respect to the distance $dist(V, V') = \sup_x |V(x) - V'(x)|$.*

**Lemma D.2.** *The $\bar{w}_h$ in line 5 of Algorithm 3 is always bounded by $\|\bar{w}_h\|_2 \leq H\sqrt{dN}$.*

*Proof of Lemma D.2.* For any $\theta \in \mathbb{R}^d$ with $\|\theta\|_2 = 1$, we have

$$
\begin{aligned}
|\theta^\top \bar{w}_h| =& |\theta^\top (\Lambda_h)^{-1} \sum_{n=1}^N \phi(s_h^n, a_h^n) V_{h+1}(s_{h+1}^n)| \\
\leq& \sum_{n=1}^N |\theta^\top (\Lambda_h)^{-1} \phi(s_h^n, a_h^n)| \cdot H \\
\leq& H \cdot \sqrt{[\sum_{n=1}^N \theta^\top (\Lambda_h)^{-1} \theta] \cdot [\sum_{n=1}^N \phi(s_h^n, a_h^n)^\top (\Lambda_h)^{-1} \phi(s_h, a_h)]} \\
\leq& H\sqrt{dN}.
\end{aligned}
$$

The second inequality is because of Cauchy-Schwarz inequality. The last inequality holds according to Lemma D.1 of Jin et al. [2020b]. $\square$

## D.3 Upper bound of estimation error

We first consider the covering number of $V_h$ in Algorithm 3. All $V_h$ can be written as:

$$V_h(\cdot) = \left( \phi(\cdot, \pi_h(\cdot))^\top (\bar{w}_h + \theta_h) \right)_{[0,H]}, \tag{12}$$

where $\theta_h$ is the parameter with respect to $r_h$ ($r_h(s,a) = \langle \phi(s,a), \theta_h \rangle$).

Note that $\Pi_{\epsilon_0,h}^{eval} \times \mathcal{W}_\epsilon$ (where $\mathcal{W}_\epsilon$ is $\epsilon$-cover of $\mathcal{B}^d(2H\sqrt{dN})$) provides a $\epsilon$-cover of $\{V_h\}$. Therefore, the $\epsilon$-covering number $\mathcal{N}_\epsilon$ of $\{V_h\}$ is bounded by

$$\log \mathcal{N}_\epsilon \leq \log |\Pi_{\epsilon_0,h}^{eval}| + \log |\mathcal{W}_\epsilon| \leq d \log(1 + \frac{8H^2\sqrt{d}}{\epsilon_0}) + d \log(1 + \frac{4H\sqrt{dN}}{\epsilon}). \tag{13}$$

Now we have the following key lemma.

**Lemma D.3.** *With probability $1 - \delta$, for any policy $\pi \in \Pi_{\epsilon_0}^{eval}$ and any linear reward function $r$ that may appear in Algorithm 3, the $\{V_h\}_{h \in [H]}$ derived by Algorithm 3 satisfies that for any $h \in [H]$,*

$$\left\| \sum_{n=1}^N \phi_h^n \left( V_{h+1}(s_{h+1}^n) - \sum_{s' \in \mathcal{S}} P_h(s'|s_h^n, a_h^n) V_{h+1}(s') \right) \right\|_{\Lambda_h^{-1}} \leq cH\sqrt{d} \cdot \sqrt{\log(\frac{Hd}{\epsilon_0\delta}) + \log(\frac{N}{\delta})},$$

*for some universal constant $c > 0$.*

*Proof of Lemma D.3.* The proof is by plugging $\epsilon = \frac{H\sqrt{d}}{N}$ in Lemma D.1 and using (13). □

**Remark D.4.** *Assume the final goal is to find $\epsilon$-optimal policy for all reward functions, we can choose that $\epsilon_0 \geq poly(\epsilon)$ and $N \leq poly(d, H, \frac{1}{\epsilon})$. Then the R.H.S. of Lemma D.3 is of order $\widetilde{O}(H\sqrt{d})$, which effectively saves a factor of $\sqrt{d}$ compared to Lemma A.1 of Wang et al. [2020].*

Now we are ready to prove the following lemma.

**Lemma D.5.** *With probability $1 - \delta$, for any policy $\pi \in \Pi_{\epsilon_0}^{eval}$ and any linear reward function $r$ that may appear in Algorithm 3, the $\{V_h\}_{h\in[H]}$ and $\{\bar{w}_h\}_{h\in[H]}$ derived by Algorithm 3 satisfies that for all $h, s, a \in [H] \times \mathcal{S} \times \mathcal{A}$,*

$$|\phi(s,a)^\top \bar{w}_h - \sum_{s'\in\mathcal{S}} P_h(s'|s,a)V_{h+1}(s')| \leq c'H\sqrt{d} \cdot \sqrt{\log(\frac{Hd}{\epsilon_0\delta}) + \log(\frac{N}{\delta})} \cdot \|\phi(s,a)\|_{\Lambda_h^{-1}},$$

*for some universal constant $c' > 0$.*

This part of proof is similar to the proof of Lemma 3.1 in Wang et al. [2020]. For completeness, we state it here.

*Proof of Lemma D.5.* Since $P_h(s'|s,a) = \phi(s,a)^\top \mu_h(s')$, we have

$$\sum_{s'\in\mathcal{S}} P_h(s'|s,a)V_{h+1}(s') = \phi(s,a)^\top \widetilde{w}_h,$$

for some $\|\widetilde{w}_h\|_2 \leq H\sqrt{d}$. Therefore, we have

$$\begin{aligned}
&\phi(s,a)^\top \bar{w}_h - \sum_{s'\in\mathcal{S}} P_h(s'|s,a)V_{h+1}(s') \\
=&\phi(s,a)^\top(\Lambda_h)^{-1}\sum_{n=1}^N \phi_h^n \cdot V_{h+1}(s_{h+1}^n) - \sum_{s'\in\mathcal{S}} P_h(s'|s,a)V_{h+1}(s') \\
=&\phi(s,a)^\top(\Lambda_h)^{-1}\left(\sum_{n=1}^N \phi_h^n \cdot V_{h+1}(s_{h+1}^n) - \Lambda_h\widetilde{w}_h\right) \\
=&\phi(s,a)^\top(\Lambda_h)^{-1}\left(\sum_{n=1}^N \phi_h^n V_{h+1}(s_{h+1}^n) - \widetilde{w}_h - \sum_{n=1}^N \phi_h^n(\phi_h^n)^\top \widetilde{w}_h\right) \\
=&\phi(s,a)^\top(\Lambda_h)^{-1}\left(\sum_{n=1}^N \phi_h^n\left(V_{h+1}(s_{h+1}^n) - \sum_{s'} P_h(s'|s_h^n, a_h^n)V_{h+1}(s')\right) - \widetilde{w}_h\right).
\end{aligned} \tag{14}$$

It holds that,

$$\begin{aligned}
&\left|\phi(s,a)^\top(\Lambda_h)^{-1}\left(\sum_{n=1}^N \phi_h^n\left(V_{h+1}(s_{h+1}^n) - \sum_{s'} P_h(s'|s_h^n, a_h^n)V_{h+1}(s')\right)\right)\right| \\
\leq&\|\phi(s,a)\|_{\Lambda_h^{-1}} \cdot \left\|\sum_{n=1}^N \phi_h^n\left(V_{h+1}(s_{h+1}^n) - \sum_{s'\in\mathcal{S}} P_h(s'|s_h^n, a_h^n)V_{h+1}(s')\right)\right\|_{\Lambda_h^{-1}} \\
\leq&cH\sqrt{d} \cdot \sqrt{\log(\frac{Hd}{\epsilon_0\delta}) + \log(\frac{N}{\delta})} \cdot \|\phi(s,a)\|_{\Lambda_h^{-1}},
\end{aligned} \tag{15}$$

for some constant $c$ due to Lemma D.3. In addition, we have

$$|\phi(s,a)^\top(\Lambda_h)^{-1}\widetilde{w}_h| \leq \|\phi(s,a)\|_{\Lambda_h^{-1}} \cdot \|\widetilde{w}_h\|_{\Lambda_h^{-1}} \leq H\sqrt{d} \cdot \|\phi(s,a)\|_{\Lambda_h^{-1}}.$$

Combining these two results, we have

$$|\phi(s,a)^\top \bar{w}_h - \sum_{s' \in \mathcal{S}} P_h(s'|s,a)V_{h+1}(s')| \leq c'H\sqrt{d} \cdot \sqrt{\log(\frac{Hd}{\epsilon_0 \delta}) + \log(\frac{N}{\delta})} \cdot \|\phi(s,a)\|_{\Lambda_h^{-1}}.$$

$\square$

Finally, the error bound of our estimations are summarized in the following lemma.

**Lemma D.6.** *For $\pi \in \Pi_{\epsilon_0}^{eval}$ and linear reward function $r$, let the output of Algorithm 3 be $\widehat{V}^\pi(r)$. Then with probability $1 - \delta$, for any policy $\pi \in \Pi_{\epsilon_0}^{eval}$ and any linear reward function $r$, it holds that*

$$|\widehat{V}^\pi(r) - V^\pi(r)| \leq c'H\sqrt{d} \cdot \sqrt{\log(\frac{Hd}{\epsilon_0 \delta}) + \log(\frac{N}{\delta})} \cdot \mathbb{E}_\pi \sum_{h=1}^{H} \|\phi(s_h, a_h)\|_{\Lambda_h^{-1}}, \qquad (16)$$

*for some universal constant $c' > 0$.*

*Proof of Lemma D.6.* For any policy $\pi \in \Pi_{\epsilon_0}^{eval}$ and any linear reward function $r$, consider the $V_h$ functions and $\bar{w}_h$ in Algorithm 3, we have

$$|V_1(s_1) - V_1^\pi(s_1)| \leq \mathbb{E}_\pi \left| \phi(s_1, a_1)^\top \bar{w}_1 + r_1(s_1, a_1) - \sum_{s' \in \mathcal{S}} P_1(s'|s_1, a_1)V_2^\pi(s') - r_1(s_1, a_1) \right|$$

$$\leq \mathbb{E}_\pi \left| \phi(s_1, a_1)^\top \bar{w}_1 - \sum_{s' \in \mathcal{S}} P_1(s'|s_1, a_1)V_2(s') \right| + \mathbb{E}_\pi \sum_{s' \in \mathcal{S}} P_1(s'|s_1, a_1) |V_2(s') - V_2^\pi(s')|$$

$$\leq \mathbb{E}_\pi c'H\sqrt{d} \cdot \sqrt{\log(\frac{Hd}{\epsilon_0 \delta}) + \log(\frac{N}{\delta})} \cdot \|\phi(s_1, a_1)\|_{\Lambda_1^{-1}} + \mathbb{E}_\pi |V_2(s_2) - V_2^\pi(s_2)|$$

$$\leq \cdots$$

$$\leq c'H\sqrt{d} \cdot \sqrt{\log(\frac{Hd}{\epsilon_0 \delta}) + \log(\frac{N}{\delta})} \cdot \mathbb{E}_\pi \sum_{h=1}^{H} \|\phi(s_h, a_h)\|_{\Lambda_h^{-1}},$$

$$(17)$$

where the first inequality results from the fact that $V_1^\pi(s_1) \in [0, H]$. The third inequality comes from Lemma D.5. The forth inequality is due to recursive application of decomposition. $\square$

**Remark D.7.** *Compared to the analysis in Wang et al. [2020] and Huang et al. [2022], our analysis saves a factor of $\sqrt{d}$. This is achieved by discretization of the policy set and bypassing the need to cover the quadratic bonus term. More specifically, the log-covering number of our $\Pi_h^{eval}$ is $\widetilde{O}(d)$. Combining with the covering set of Euclidean ball in $\mathbb{R}^d$, the total log-covering number is still $\widetilde{O}(d)$. In contrast, both previous works need to cover bonus like $\sqrt{\phi(\cdot, \cdot)^\top (\Lambda)^{-1} \phi(\cdot, \cdot)}$, which requires the log-covering number to be $\widetilde{O}(d^2)$.*

# E  Generalized algorithms for estimating value functions

Since $\Pi^{exp}$ we construct in Section C.3 is guaranteed to cover explorative policies under any feasible linear MDP, it suffices to do exploration using only policies from $\Pi^{exp}$. In this section, we generalize the algorithm we propose in Section D for our purpose during exploration phase. To be more specific, we design an algorithm to estimate $\mathbb{E}_\pi r(s_h, a_h)$ for any policy $\pi \in \Pi^{exp}$ and any reward $r$. Recall that given accuracy $\epsilon_1$, the policy set we construct in Section C.3 is $\Pi_{\epsilon_1}^{exp}$ and the policy set for layer $h$ is $\Pi_{\epsilon_1, h}^{exp}$.

## E.1 The algorithm

---

**Algorithm 4** Estimation of $\mathbb{E}_\pi r(s_h, a_h)$ given exploration data (EstimateER)

1: **Input:** Policy to evaluate $\pi \in \Pi_{\epsilon_1}^{exp}$. Reward function $r(s,a)$ and its uniform upper bound $A$. Layer $h$. Exploration data $\{s_{\widetilde{h}}^n, a_{\widetilde{h}}^n\}_{(\widetilde{h},n) \in [H] \times [N]}$. Initial state $s_1$.
2: **Initialization:** $Q_h(\cdot, \cdot) \leftarrow r(\cdot, \cdot)$, $V_h(\cdot) \leftarrow Q_h(\cdot, \pi_h(\cdot))$.
3: **for** $\widetilde{h} = h-1, h-2, \ldots, 1$ **do**
4: $\quad \Lambda_{\widetilde{h}} \leftarrow I + \sum_{n=1}^N \phi(s_{\widetilde{h}}^n, a_{\widetilde{h}}^n)\phi(s_{\widetilde{h}}^n, a_{\widetilde{h}}^n)^\top$.
5: $\quad \bar{w}_{\widetilde{h}} \leftarrow (\Lambda_{\widetilde{h}})^{-1} \sum_{n=1}^N \phi(s_{\widetilde{h}}^n, a_{\widetilde{h}}^n) V_{\widetilde{h}+1}(s_{\widetilde{h}+1}^n)$.
6: $\quad Q_{\widetilde{h}}(\cdot, \cdot) \leftarrow (\phi(\cdot, \cdot)^\top \bar{w}_{\widetilde{h}})_{[0,A]}$.
7: $\quad V_{\widetilde{h}}(\cdot) \leftarrow Q_{\widetilde{h}}(\cdot, \pi_{\widetilde{h}}(\cdot))$.
8: **end for**
9: **Output:** $V_1(s_1)$.

---

Algorithm 4 applies LSVI to estimate $\mathbb{E}_\pi r(s_h, a_h)$ for any $\pi \in \Pi_{\epsilon_1}^{exp}$ (according to our construction, all possible $\pi$'s are deterministic), any reward function $r$ and any time step $h$. Note that the algorithm takes the uniform upper bound $A$ of all possible reward functions (i.e., for any reward function $r$ that may appear as the input, $r \in [0, A]$) as the input, and uses the value of $A$ to truncate the Q-function in line 6. Algorithm 4 looks similar to Algorithm 3 while there are two key differences. First, the reward function is non-zero at only one layer in Algorithm 4 while the reward function in Algorithm 3 can be any valid reward functions. In addition, Algorithm 4 takes the upper bound of reward function as input and uses this value to bound the Q-functions while Algorithm 3 uses $H$ as the upper bound.

## E.2 Technical Lemmas

**Lemma E.1** (Generalization of Lemma D.4 of Jin et al. [2020b]). *Let $\{x_\tau\}_{\tau=1}^\infty$ be a stochastic process on state space $\mathcal{S}$ with corresponding filtration $\{\mathcal{F}_\tau\}_{\tau=0}^\infty$. Let $\{\phi_\tau\}_{\tau=1}^\infty$ be an $\mathbb{R}^d$-valued stochastic process where $\phi_\tau \in \mathcal{F}_{\tau-1}$, and $\|\phi_\tau\| \leq 1$. Let $\Lambda_k = I + \sum_{\tau=1}^k \phi_\tau \phi_\tau^\top$. Then for any $\delta > 0$, with probability at least $1 - \delta$, for all $k \geq 0$, and any $V \in \mathcal{V}$ so that $\sup_x |V(x)| \leq A$, we have:*

$$\left\| \sum_{\tau=1}^k \phi_\tau \{V(x_\tau) - \mathbb{E}[V(x_\tau)|\mathcal{F}_{\tau-1}]\} \right\|_{\Lambda_k^{-1}}^2 \leq 4A^2 \left[ \frac{d}{2}\log(k+1) + \log(\frac{\mathcal{N}_\epsilon}{\delta}) \right] + 8k^2\epsilon^2,$$

*where $\mathcal{N}_\epsilon$ is the $\epsilon$-covering number of $\mathcal{V}$ with respect to the distance $dist(V, V') = \sup_x |V(x) - V'(x)|$.*

**Lemma E.2.** *If $A \leq 1$, the $\bar{w}_{\widetilde{h}}$ in line 5 of Algorithm 4 is always bounded by $\|\bar{w}_{\widetilde{h}}\|_2 \leq \sqrt{dN}$.*

*Proof of Lemma E.2.* The proof is almost identical to Lemma D.2, the only difference is that $H$ is replaced by 1. $\qquad\square$

## E.3 Upper bound of estimation error

We first consider the covering number of all possible $V_h$ in Algorithm 4. In the remaining part of this section, we assume that the set of all reward functions to be estimated is $\bar{\mathcal{R}}$ with uniform upper bound $A_{\bar{\mathcal{R}}} \leq 1$. In addition, assume there exists $\epsilon$-covering $\bar{\mathcal{R}}_\epsilon$ of $\bar{\mathcal{R}}$ with covering number $\log(|\bar{\mathcal{R}}_\epsilon|) = B_\epsilon$.[13]

For fixed $h \in [H]$, under the case where the layer to estimate is exactly $h$, $V_h$ can be written as:

$$V_h(\cdot) = r(\cdot, \pi_h(\cdot)). \tag{18}$$

The set $\Pi_{\epsilon_1,h}^{exp} \times \bar{\mathcal{R}}_\epsilon$ provides an $\epsilon$-covering of $V_h$. Thus the covering number under this case is $|\Pi_{\epsilon_1,h}^{exp}| \cdot |\bar{\mathcal{R}}_\epsilon|$.

---

[13]We will show that all cases we consider in this paper satisfy these two assumptions.

In addition, if the layer to estimate is some $h' > h$, then $V_h$ can be written as:

$$V_h(\cdot) = (\phi(\cdot, \pi_h(\cdot))^\top \bar{w}_h)_{[0, A_{\bar{\mathcal{R}}}]}, \tag{19}$$

where the set $\Pi_{\epsilon_1, h}^{exp} \times \mathcal{W}_\epsilon$ ($\mathcal{W}_\epsilon$ is $\epsilon$-covering of $\mathcal{B}^d(\sqrt{dN})$) provides an $\epsilon$-covering of $V_h$. The covering number under this case is $|\Pi_{\epsilon_1, h}^{exp}| \cdot |\mathcal{W}_\epsilon|$.

Since all possible $V_h$ is either the case in (18) (the layer to estimate is exactly $h$) or (19) (the layer to estimate is larger than $h$), for any $h \in [H]$ the $\epsilon$-covering number $\mathcal{N}_\epsilon$ of all possible $V_h$ satisfies that:

$$\begin{aligned}
\log \mathcal{N}_\epsilon &\leq \log(|\Pi_{\epsilon_1, h}^{exp}| \cdot |\bar{\mathcal{R}}_\epsilon| + |\Pi_{\epsilon_1, h}^{exp}| \cdot |\mathcal{W}_\epsilon|) \\
&\leq \log(|\Pi_{\epsilon_1, h}^{exp}|) + \log(|\bar{\mathcal{R}}_\epsilon|) + \log(|\mathcal{W}_\epsilon|) \\
&\leq 2d^2 \log(1 + \frac{32H^2\sqrt{d}}{\epsilon_1^2}) + d\log(1 + \frac{2\sqrt{dN}}{\epsilon}) + B_\epsilon.
\end{aligned} \tag{20}$$

Now we have the following key lemma. The proof is almost identical to Lemma D.3, so we omit it here.

**Lemma E.3.** *With probability $1 - \delta$, for any policy $\pi \in \Pi_{\epsilon_1}^{exp}$, any reward function $r \in \bar{\mathcal{R}}$ that may appear in Algorithm 4 (with the input $A = A_{\bar{\mathcal{R}}}$) and layer $h$, the $\{V_{\tilde{h}}\}_{\tilde{h} \in [h]}$ derived by Algorithm 4 satisfies that for any $\tilde{h} \in [h-1]$,*

$$\begin{aligned}
&\left\| \sum_{n=1}^N \phi_{\tilde{h}}^n \left( V_{\tilde{h}+1}(s_{\tilde{h}+1}^n) - \sum_{s' \in \mathcal{S}} P_{\tilde{h}}(s'|s_{\tilde{h}}^n, a_{\tilde{h}}^n) V_{\tilde{h}+1}(s') \right) \right\|_{\Lambda_{\tilde{h}}^{-1}} \\
&\leq c A_{\bar{\mathcal{R}}} \cdot \sqrt{d^2 \log(\frac{Hd}{\epsilon_1 \delta}) + d\log(\frac{N}{\delta}) + B_{A_{\bar{\mathcal{R}}}/N} + \log(\frac{1}{\delta})},
\end{aligned} \tag{21}$$

*for some universal constant $c > 0$.*

Now we can provide the following Lemma E.4 whose proof is almost identical to Lemma D.5. The only difference is that $H$ is replaced by $A_{\bar{\mathcal{R}}}$.

**Lemma E.4.** *With probability $1 - \delta$, for any policy $\pi \in \Pi_{\epsilon_1}^{exp}$, any reward function $r \in \bar{\mathcal{R}}$ that may appear in Algorithm 4 (with the input $A = A_{\bar{\mathcal{R}}}$) and layer $h$, the $\{V_{\tilde{h}}\}_{\tilde{h} \in [h]}$ and $\{\bar{w}_{\tilde{h}}\}_{\tilde{h} \in [h-1]}$ derived by Algorithm 4 satisfies that for all $\tilde{h}, s, a \in [h-1] \times \mathcal{S} \times \mathcal{A}$,*

$$\begin{aligned}
&|\phi(s, a)^\top \bar{w}_{\tilde{h}} - \sum_{s' \in \mathcal{S}} P_{\tilde{h}}(s'|s, a) V_{\tilde{h}+1}(s')| \\
&\leq c' A_{\bar{\mathcal{R}}} \cdot \sqrt{d^2 \log(\frac{Hd}{\epsilon_1 \delta}) + d\log(\frac{N}{\delta}) + B_{A_{\bar{\mathcal{R}}}/N} + \log(\frac{1}{\delta})} \cdot \|\phi(s, a)\|_{\Lambda_{\tilde{h}}^{-1}},
\end{aligned} \tag{22}$$

*for some universal constant $c' > 0$.*

Finally, the error bound of our estimations are summarized in the following lemma.

**Lemma E.5.** *For any policy $\pi \in \Pi_{\epsilon_1}^{exp}$, any reward function $r \in \bar{\mathcal{R}}$ that may appear in Algorithm 4 (with the input $A = A_{\bar{\mathcal{R}}}$) and layer $h$, let the output of Algorithm 4 be $\widehat{\mathbb{E}}_\pi r(s_h, a_h)$. Then with probability $1 - \delta$, for any policy $\pi \in \Pi_{\epsilon_1}^{exp}$, any reward function $r \in \bar{\mathcal{R}}$ and any layer $h$, it holds that*

$$\begin{aligned}
&|\widehat{\mathbb{E}}_\pi r(s_h, a_h) - \mathbb{E}_\pi r(s_h, a_h)| \\
&\leq c' A_{\bar{\mathcal{R}}} \cdot \sqrt{d^2 \log(\frac{Hd}{\epsilon_1 \delta}) + d\log(\frac{N}{\delta}) + B_{A_{\bar{\mathcal{R}}}/N}} \cdot \mathbb{E}_\pi \sum_{\tilde{h}=1}^{h-1} \|\phi(s_{\tilde{h}}, a_{\tilde{h}})\|_{\Lambda_{\tilde{h}}^{-1}},
\end{aligned} \tag{23}$$

*for some universal constant $c' > 0$.*

*Proof of Lemma E.5.* For any policy $\pi \in \Pi_{\epsilon_0}^{exp}$, any reward function $r \in \bar{\mathcal{R}}$ and any layer $h$, consider the $\{V_{\tilde{h}}\}_{\tilde{h} \in [h]}$ functions and $\{\bar{w}_{\tilde{h}}\}_{\tilde{h} \in [h-1]}$ in Algorithm 4, we have $\widehat{\mathbb{E}}_\pi r(s_h, a_h) = V_1(s_1)$. Besides,

we abuse the notation and let $r$ denote the reward function where $r_{h'}(s,a) = \mathbb{1}(h' = h)r(s,a)$, let the value function under this $r$ be $V_{\widetilde{h}}^\pi(s)$, then $V_1^\pi(s_1) = \mathbb{E}_\pi r(s_h, a_h)$. It holds that

$$
\begin{aligned}
&|\widehat{\mathbb{E}}_\pi r(s_h, a_h) - \mathbb{E}_\pi r(s_h, a_h)| \\
=&|V_1(s_1) - V_1^\pi(s_1)| \\
\leq&\mathbb{E}_\pi \left| \phi(s_1, a_1)^\top \bar{w}_1 - \sum_{s' \in \mathcal{S}} P_1(s'|s_1, a_1) V_2^\pi(s') \right| \\
\leq&\mathbb{E}_\pi \left| \phi(s_1, a_1)^\top \bar{w}_1 - \sum_{s' \in \mathcal{S}} P_1(s'|s_1, a_1) V_2(s') \right| + \mathbb{E}_\pi \sum_{s' \in \mathcal{S}} P_1(s'|s_1, a_1) |V_2(s') - V_2^\pi(s')| \\
\leq&\mathbb{E}_\pi c' A_{\overline{\mathcal{R}}} \cdot \sqrt{d^2 \log(\frac{Hd}{\epsilon_1 \delta}) + d \log(\frac{N}{\delta}) + B_{A_{\overline{\mathcal{R}}}/N}} \cdot \|\phi(s_1, a_1)\|_{\Lambda_1^{-1}} + \mathbb{E}_\pi |V_2(s_2) - V_2^\pi(s_2)| \\
\leq&\cdots \\
\leq&c' A_{\overline{\mathcal{R}}} \cdot \sqrt{d^2 \log\frac{Hd}{\epsilon_1 \delta} + d\log(\frac{N}{\delta}) + B_{A_{\overline{\mathcal{R}}}/N}} \cdot \mathbb{E}_\pi \sum_{\widetilde{h}=1}^{h-1} \|\phi(s_{\widetilde{h}}, a_{\widetilde{h}})\|_{\Lambda_{\widetilde{h}}^{-1}} + \mathbb{E}_\pi |V_h(s_h) - V_h^\pi(s_h)| \\
=&c' A_{\overline{\mathcal{R}}} \cdot \sqrt{d^2 \log(\frac{Hd}{\epsilon_1 \delta}) + d\log(\frac{N}{\delta}) + B_{A_{\overline{\mathcal{R}}}/N}} \cdot \mathbb{E}_\pi \sum_{\widetilde{h}=1}^{h-1} \|\phi(s_{\widetilde{h}}, a_{\widetilde{h}})\|_{\Lambda_{\widetilde{h}}^{-1}},
\end{aligned}
\tag{24}
$$

where the first inequality results from the fact that $V_1^\pi(s_1) \in [0, A_{\overline{\mathcal{R}}}]$. The third inequality comes from Lemma E.4. The fifth inequality is due to recursive application of decomposition. The last equation holds since $V_h(\cdot) = V_h^\pi(\cdot) = r(\cdot, \pi_h(\cdot))$. $\qquad\square$

**Remark E.6.** *From Lemma E.5, we can see that the estimation error at layer $h$ can be bounded by the summation of uncertainty from the previous layers, with additional factor of $\widetilde{O}(Ad)$. Therefore, if the uncertainty of all previous layers are small with respect to $\Pi^{exp}$, we can estimate $\mathbb{E}_\pi r_h$ accurately for any $\pi \in \Pi^{exp}$ and any reward $r$ from a large set of reward functions.*

**Remark E.7.** *Note that we only need to estimate $\mathbb{E}_\pi r(s_h, a_h)$ accurately for $\pi \in \Pi^{exp}$. For $\pi \in \Delta(\Pi^{exp})$, if $\pi$ takes policy $\pi_i \in \Pi^{exp}$ with probability $p_i$ (for $i \in [k]$), then we define*

$$
\widehat{\mathbb{E}}_\pi r(s_h, a_h) := \sum_{i \in [k]} p_i \cdot \widehat{\mathbb{E}}_{\pi_i} r(s_h, a_h),
\tag{25}
$$

*where $\widehat{\mathbb{E}}_\pi r(s_h, a_h)$ is the estimation we acquire w.r.t policy $\pi$ and $\widehat{\mathbb{E}}_{\pi_i} r(s_h, a_h)$ is the output of Algorithm 4 with input $\pi_i \in \Pi^{exp}$. Assume that for all $\pi \in \Pi^{exp}$, $|\widehat{\mathbb{E}}_\pi r(s_h, a_h) - \mathbb{E}_\pi r(s_h, a_h)| \leq e$, we have for all $\pi \in \Delta(\Pi^{exp})$, $|\widehat{\mathbb{E}}_\pi r(s_h, a_h) - \mathbb{E}_\pi r(s_h, a_h)| \leq \sum_i p_i |\widehat{\mathbb{E}}_{\pi_i} r(s_h, a_h) - \mathbb{E}_{\pi_i} r(s_h, a_h)| \leq e$. Therefore, the conclusion of Lemma E.5 naturally holds for $\pi \in \Delta(\Pi^{exp})$.*

# F  Proof of Theorem 5.1

Recall that $\iota = \log(dH/\epsilon\delta)$, $\bar{\epsilon} = \frac{C_1 \epsilon}{H^2 \sqrt{d \cdot \iota}}$. The explorative policy set we construct is $\Pi_{\frac{\epsilon}{3}}^{exp}$ while the policies to evaluate is $\Pi_{\frac{\epsilon}{3}}^{eval}$. Number of episodes for each deployment is $N = \frac{C_2 d\iota}{\bar{\epsilon}^2} = \frac{C_2 d^2 H^4 \iota^3}{C_1^2 \epsilon^2}$. In addition, $\Sigma_\pi$ is short for $\mathbb{E}_\pi[\phi_h \phi_h^\top]$ while $\widehat{\Sigma}_\pi$ is short for $\widehat{\mathbb{E}}_\pi[\phi_h \phi_h^\top]$. For clarity, we restrict our choice that $0 < C_1 < 1$ and $C_2, C_3 > 1$. We begin with detailed explanation of $\widehat{\Sigma}_\pi$ and $\widehat{\mathbb{E}}_{\widehat{\pi}} \left[ \phi(s_h, a_h)^\top (N \cdot \widehat{\Sigma}_\pi)^{-1} \phi(s_h, a_h) \right]$ from (1).

## F.1  Detailed explanation

First of all, as have been pointed out in Algorithm 1, $\widehat{\Sigma}_\pi$ is short for $\widehat{\mathbb{E}}_\pi[\phi(s_h, a_h)\phi(s_h, a_h)^\top]$. Assume the feature map is $\phi(s,a) = (\phi_1(s,a), \phi_2(s,a), \cdots, \phi_d(s,a))^\top$, where $\phi_i(s,a) \in \mathbb{R}$. Then the estimation of covariance matrix is calculated pointwisely. For each coordinate $i, j \in [d] \times [d]$,

we use Algorithm 4 to estimate $\mathbb{E}_\pi r(s_h, a_h) = \mathbb{E}_\pi \frac{\phi_i(s_h,a_h)\phi_j(s_h,a_h)+1}{2}$ [14]. More specifically, for any $\pi \in \Pi_{\frac{\epsilon}{3}}^{exp}$, $\widehat{\Sigma}_{\pi(ij)} = 2\widehat{E}_{ij} - 1$, where $\widehat{E}_{ij}$ is the output of Algorithm 4 with input $\pi$, $r(s,a) = \frac{\phi_i(s,a)\phi_j(s,a)+1}{2}$ with $A = 1$, layer $h$ and exploration dataset $\mathcal{D}$. Therefore, the set of all possible rewards is $\bar{\mathcal{R}} = \{\frac{\phi_i(s,a)\phi_j(s,a)+1}{2}, (i,j) \in [d] \times [d]\}$. The set $\bar{\mathcal{R}}$ is a covering set of itself with log-covering number $B_\epsilon = \log(|\bar{\mathcal{R}}|) = 2\log d$. In addition, note that the estimation $\widehat{\Sigma}_{\pi(ij)} = \widehat{\Sigma}_{\pi(ji)}$ for all $i, j$, which means the estimation $\widehat{\Sigma}_\pi$ is symmetric. The above discussion tackles the case where $\pi \in \Pi_{\frac{\epsilon}{3}}^{exp}$, for the general case where $\pi \in \Delta(\Pi_{\frac{\epsilon}{3}}^{exp})$, the estimation is derived by (25) in Remark E.7. In the discussion below, we only need to bound $\|\widehat{\mathbb{E}}_\pi \phi_h \phi_h^\top - \mathbb{E}_\pi \phi_h \phi_h^\top\|_2$ for all $\pi \in \Pi_{\frac{\epsilon}{3}}^{exp}$ and the same bound applies to all $\pi \in \Delta(\Pi_{\frac{\epsilon}{3}}^{exp})$.

The second estimator is $\widehat{\mathbb{E}}_{\widehat{\pi}}\left[\phi(s_h, a_h)^\top (N \cdot \widehat{\Sigma}_\pi)^{-1} \phi(s_h, a_h)\right]$, which is calculated via directly applying Algorithm 4 with input $\widehat{\pi} \in \Pi_{\frac{\epsilon}{3}}^{exp}$, $r(s,a) = \phi(s,a)^\top (N \cdot \widehat{\Sigma}_\pi)^{-1} \phi(s,a)$ with $A = \frac{\bar{\epsilon}}{C_2 d^3 H \iota^2} = \frac{C_1 \epsilon}{C_2 d^{7/2} H^3 \iota^3}$, layer $h$ and exploration dataset $\mathcal{D}$. Note that the validity of uniform upper bound $A$ holds since we only consider the case where $\lambda_{\min}(\widehat{\Sigma}_\pi) \geq d^2 H \bar{\epsilon} \iota$, which means that $\lambda_{\min}(N \cdot \widehat{\Sigma}_\pi) \geq d^2 H \bar{\epsilon} \iota \cdot \frac{C_2 d \iota}{\bar{\epsilon}^2} = \frac{C_2 d^3 H \iota^2}{\bar{\epsilon}}$. Therefore the set of all possible rewards is subset of $\bar{\mathcal{R}} = \{r(s,a) = \phi(s,a)^\top (\Sigma)^{-1} \phi(s,a) | \lambda_{\min}(\Sigma) \geq \frac{C_2 d^{7/2} H^3 \iota^3}{C_1 \epsilon}\}$ and the $\epsilon$-covering number is characterized by Lemma F.3 below.

## F.2 Technical lemmas

In this part, we state some technical lemmas.

**Lemma F.1** (Lemma H.4 of Min et al. [2021]). *Let $\phi : \mathcal{S} \times \mathcal{A} \to \mathbb{R}^d$ satisfies $\|\phi(s,a)\| \leq C$ for all $s, a \in \mathcal{S} \times \mathcal{A}$. For any $K > 0, \lambda > 0$, define $\bar{G}_K = \sum_{k=1}^{K} \phi(s_k, a_k)\phi(s_k, a_k)^\top + \lambda I_d$ where $(s_k, a_k)$'s are i.i.d samples from some distribution $\nu$. Then with probability $1 - \delta$,*

$$\left\|\frac{\bar{G}_K}{K} - \mathbb{E}_\nu\left[\frac{\bar{G}_K}{K}\right]\right\|_2 \leq \frac{4\sqrt{2}C^2}{\sqrt{K}}\left(\log\frac{2d}{\delta}\right)^{1/2}. \tag{26}$$

**Lemma F.2** (Corollary of Lemma D.6). *There exists universal constant $c_D > 0$, such that with our choice of $\epsilon_0 = \frac{\epsilon}{3}$ and $N = \frac{C_2 d^2 H^4 \iota^3}{C_1^2 \epsilon^2}$, the multiplicative factor of (16) satisfies that*

$$c' H \sqrt{d} \cdot \sqrt{\log(\frac{Hd}{\epsilon_0 \delta}) + \log(\frac{N}{\delta})} \leq c_D H \sqrt{d} \cdot \log(\frac{C_2 dH}{C_1 \epsilon \delta}). \tag{27}$$

*Proof of Lemma F.2.* The existence of universal constant $c_D$ holds since $c'$ in (16) is universal constant and direct calculation. $\square$

**Lemma F.3** (Covering number). *Consider the set of possible rewards $\bar{\mathcal{R}} = \{r(s,a) = \phi(s,a)^\top (\Sigma)^{-1} \phi(s,a) | \lambda_{\min}(\Sigma) \geq \frac{C_2 d^{7/2} H^3 \iota^3}{C_1 \epsilon}\}$. Let $A_{\bar{\mathcal{R}}} = \frac{C_1 \epsilon}{C_2 d^{7/2} H^3 \iota^3}$ and $N = \frac{C_2 d^2 H^4 \iota^3}{C_1^2 \epsilon^2}$, we have that the $\frac{A_{\bar{\mathcal{R}}}}{N}$-cover $\mathcal{R}_{A_{\bar{\mathcal{R}}}/N}$ of $\bar{\mathcal{R}}$ satisfies that for some universal constant $c_F > 0$,*

$$B_{A_{\bar{\mathcal{R}}}/N} = \log(|\bar{\mathcal{R}}_{A_{\bar{\mathcal{R}}}/N}|) \leq c_F d^2 \log(\frac{C_2 dH}{C_1 \epsilon}). \tag{28}$$

*Proof of Lemma F.3.* The conclusion holds due to Lemma D.6 of Jin et al. [2020b] and direct calculation. $\square$

**Lemma F.4** (Corollary of Lemma E.5). *There exists universal constant $c_E^1 > 0$ such that for the first case in Section F.1 with our choice of $\epsilon_1 = \frac{\epsilon}{3}$, $A = 1$, $B = 2\log(d)$ and $N = \frac{C_2 d^2 H^4 \iota^3}{C_1^2 \epsilon^2}$, the multiplicative factor of (23) satisfies that*

$$c' A_{\bar{\mathcal{R}}} \cdot \sqrt{d^2 \log(\frac{Hd}{\epsilon_1 \delta}) + d\log(\frac{N}{\delta}) + B_{A_{\bar{\mathcal{R}}}/N}} \leq c_E^1 \cdot d\log(\frac{C_2 dH}{C_1 \epsilon \delta}). \tag{29}$$

---

[14]The transformation is to ensure that the reward is larger than 0.

*Proof of Lemma F.4.* The existence of universal constant $c_E^1$ holds since $c'$ in (23) is universal constant and direct calculation. $\qquad\square$

**Lemma F.5** (Corollary of Lemma E.5)**.** *There exists universal constant $c_E^2 > 0$ such that for the second case in Section F.1 with our choice of $\epsilon_1 = \frac{\epsilon}{3}$, $A = \frac{\bar{\epsilon}}{C_2 d^3 H \iota^2} = \frac{C_1 \epsilon}{C_2 d^{7/2} H^3 \iota^3}$, $B = c_F d^2 \log(\frac{C_2 dH}{C_1 \epsilon})$ and $N = \frac{C_2 d^2 H^4 \iota^3}{C_1^2 \epsilon^2}$, the multiplicative factor of (23) satisfies that*

$$c' A_{\bar{\mathcal{R}}} \cdot \sqrt{d^2 \log(\frac{Hd}{\epsilon_1 \delta}) + d \log(\frac{N}{\delta}) + B_{A_{\bar{\mathcal{R}}}/N}} \le c_E^2 \cdot \frac{\bar{\epsilon}}{C_2 d^2 H \iota} \log(\frac{C_2 dH}{C_1 \epsilon \delta}). \tag{30}$$

*Proof of Lemma F.5.* The existence of universal constant $c_E^2$ holds since $c'$ in (23) is universal constant and direct calculation. $\qquad\square$

Now that we have the universal constants $c_D, c_F, c_E^1, c_E^2$, for notational simplicity, we let $c_E = \max\{c_E^1, c_E^2\}$. Therefore, the conclusions of Lemma F.4 and F.5 hold if we replace $c_E^i$ with $c_E$.

### F.3 Choice of universal constants

In this section, we determine the choice of universal constants in Algorithm 1 and Theorem 5.1. First, $C_1, C_2$ satisfies that $C_1 \cdot C_2 = 1$, $0 < C_1 < 1$ and the following conditions:

$$c_D H \sqrt{d} \cdot \log(\frac{C_2 dH}{C_1 \epsilon \delta}) \le \frac{1}{3C_1} H \sqrt{d} \log(\frac{dH}{\epsilon \delta}). \tag{31}$$

$$c_E \cdot \frac{\bar{\epsilon}}{C_2 d^2 H \iota} \log(\frac{C_2 dH}{C_1 \epsilon \delta}) \le \frac{\bar{\epsilon}}{2 d^2 H}. \tag{32}$$

It is clear that when $C_2$ is larger than some universal threshold and $C_1 = \frac{1}{C_2}$, the constants $C_1, C_2$ satisfy the previous four conditions.

Next, we choose $C_3$ such that

$$\frac{C_3}{4} \log(\frac{dH}{\epsilon \delta}) \ge c_E \log(\frac{C_2 dH}{C_1 \epsilon \delta}), \tag{33}$$

and $C_4 = 80 C_1 C_3$. Since $c_D, c_E, c_F$ are universal constants, our $C_1, C_2, C_3, C_4$ are also universal constants that are independent with the parameters $d, H, \epsilon, \delta$.

### F.4 Restate Theorem 5.1 and our induction

**Theorem F.6** (Restate Theorem 5.1)**.** *We run Algorithm 1 to collect data and let Planning$(\cdot)$ denote the output of Algorithm 2. For the universal constants $C_1, C_2, C_3, C_4$ we choose, for any $\epsilon > 0$ and $\delta > 0$, as well as $\epsilon < \frac{H(\lambda^\star)^2}{C_4 d^{7/2} \log(1/\lambda^\star)}$, with probability $1 - \delta$, for any feasible linear reward function $r$, Planning$(r)$ returns a policy that is $\epsilon$-optimal with respect to $r$.*

Throughout the proof in this section, we assume that the condition $\epsilon < \frac{H(\lambda^\star)^2}{C_4 d^{7/2} \log(1/\lambda^\star)}$ holds. Then we state our induction condition.

**Condition F.7** (Induction Condition)**.** *Suppose after $h - 1$ deployments (i.e., after the exploration of the first $h - 1$ layers), the dataset $\mathcal{D}_{h-1} = \{s_{\tilde{h}}^n, a_{\tilde{h}}^n\}_{\tilde{h}, n \in [H] \times [(h-1)N]}$ and $\Lambda_{\tilde{h}}^{h-1} = I + \sum_{n=1}^{(h-1)N} \phi_{\tilde{h}}^n (\phi_{\tilde{h}}^n)^\top$ for all $\tilde{h} \in [H]$. The induction condition is:*

$$\max_{\pi \in \Pi_{\frac{\epsilon}{3}}^{exp}} \mathbb{E}_\pi \left[ \sum_{\tilde{h}=1}^{h-1} \sqrt{\phi(s_{\tilde{h}}, a_{\tilde{h}})^\top (\Lambda_{\tilde{h}}^{h-1})^{-1} \phi(s_{\tilde{h}}, a_{\tilde{h}})} \right] \le (h-1)\bar{\epsilon}. \tag{34}$$

Suppose that after $h$ deployments, the dataset $\mathcal{D}_h = \{s_{\widetilde{h}}^n, a_{\widetilde{h}}^n\}_{\widetilde{h}, n \in [H] \times [hN]}$ and $\Lambda_{\widetilde{h}}^h = I + \sum_{n=1}^{hN} \phi_{\widetilde{h}}^n (\phi_{\widetilde{h}}^n)^\top$ for all $\widetilde{h} \in [H]$. We will prove that given condition F.7 holds, with probability at least $1 - \delta$, the following induction holds:

$$\max_{\pi \in \Pi_{\frac{\epsilon}{3}}^{exp}} \mathbb{E}_\pi \left[ \sqrt{\phi(s_h, a_h)^\top (\Lambda_h^h)^{-1} \phi(s_h, a_h)} \right] \leq \bar{\epsilon}. \tag{35}$$

Note that the induction (35) naturally implies that

$$\max_{\pi \in \Pi_{\frac{\epsilon}{3}}^{exp}} \mathbb{E}_\pi \left[ \sum_{\widetilde{h}=1}^{h} \sqrt{\phi(s_{\widetilde{h}}, a_{\widetilde{h}})^\top (\Lambda_{\widetilde{h}}^h)^{-1} \phi(s_{\widetilde{h}}, a_{\widetilde{h}})} \right] \leq h\bar{\epsilon}. \tag{36}$$

Suppose after the whole exploration process, the dataset $\mathcal{D} = \{s_h^n, a_h^n\}_{h, n \in [H] \times [HN]}$ and $\Lambda_h = I + \sum_{n=1}^{HN} \phi_h^n (\phi_h^n)^\top$ for all $h \in [H]$. If the previous induction holds, we have with probability $1 - H\delta$,

$$\max_{\pi \in \Pi_{\frac{\epsilon}{3}}^{exp}} \mathbb{E}_\pi \left[ \sum_{h=1}^{H} \sqrt{\phi(s_h, a_h)^\top (\Lambda_h)^{-1} \phi(s_h, a_h)} \right] \leq H\bar{\epsilon}. \tag{37}$$

Next we begin the proof of such induction. We assume the Condition F.7 holds and prove (35).

## F.5 Error bound of estimation

Recall that the policy we apply to explore the $h$-th layer is

$$\pi_h = \operatorname*{argmin}_{\pi \in \Delta(\Pi_{\frac{\epsilon}{3}}^{exp}) \text{ s.t. } \lambda_{\min}(\widehat{\Sigma}_\pi) \geq C_3 d^2 H \bar{\epsilon} \iota} \max_{\widehat{\pi} \in \Pi_{\frac{\epsilon}{3}}^{exp}} \widehat{\mathbb{E}}_{\widehat{\pi}} \left[ \phi(s_h, a_h)^\top (N \cdot \widehat{\Sigma}_\pi)^{-1} \phi(s_h, a_h) \right], \tag{38}$$

where the detailed definition of $\widehat{\Sigma}_\pi$ and $\widehat{\mathbb{E}}_{\widehat{\pi}} \left[ \phi(s_h, a_h)^\top (N \cdot \widehat{\Sigma}_\pi)^{-1} \phi(s_h, a_h) \right]$ are explained in Section F.1. In addition, we define the *optimal* policy $\bar{\pi}_h^\star$ for exploring layer $h$:

$$\bar{\pi}_h^\star = \operatorname*{argmin}_{\pi \in \Delta(\Pi_{\frac{\epsilon}{3}}^{exp})} \max_{\widehat{\pi} \in \Pi_{\frac{\epsilon}{3}}^{exp}} \mathbb{E}_{\widehat{\pi}} \left[ \phi(s_h, a_h)^\top (N \cdot \Sigma_\pi)^{-1} \phi(s_h, a_h) \right], \tag{39}$$

where $\mathbb{E}_{\widehat{\pi}}$ means the actual expectation. Similarly, $\Sigma_\pi$ is short for $\mathbb{E}_\pi [\phi(s_h, a_h) \phi(s_h, a_h)^\top]$.

According to Lemma C.8, since $\epsilon \leq \frac{H(\lambda^\star)^2}{C_4 d^{7/2} \log(1/\lambda^\star)} \leq \frac{\lambda^\star}{4}$ [15], we have

$$\sup_{\pi \in \Delta(\Pi_{\frac{\epsilon}{3}}^{exp})} \lambda_{\min}(\mathbb{E}_\pi \phi_h \phi_h^\top) \geq \frac{(\lambda^\star)^2}{64 d \log(1/\lambda^\star)}. \tag{40}$$

Therefore, together with the conclusion of Lemma B.4 and our definition of $\bar{\pi}_h^\star$, it holds that:

$$\lambda_{\min}(\mathbb{E}_{\bar{\pi}_h^\star} \phi_h \phi_h^\top) \geq \frac{(\lambda^\star)^2}{64 d^2 \log(1/\lambda^\star)}. \tag{41}$$

### F.5.1 Error bound for the first estimator

We first consider the upper bound of $\left\| \widehat{\mathbb{E}}_\pi [\phi(s_h, a_h) \phi(s_h, a_h)^\top] - \mathbb{E}_\pi [\phi(s_h, a_h) \phi(s_h, a_h)^\top] \right\|_2$. Recall that (as stated in first half of Section F.1), $\widehat{\mathbb{E}}_\pi [\phi(s_h, a_h) \phi(s_h, a_h)^\top]$ is estimated through calling Algorithm 4 for each coordinate $i, j \in [d] \times [d]$. Therefore, we first bound the pointwise error.

**Lemma F.8** (Pointwise error). *With probability $1 - \delta$, for all $\pi \in \Pi_{\frac{\epsilon}{3}}^{exp}$ and all coordinates $(i, j) \in [d] \times [d]$, it holds that*

$$\left| \widehat{\mathbb{E}}_\pi [\phi(s_h, a_h) \phi(s_h, a_h)^\top]_{(ij)} - \mathbb{E}_\pi [\phi(s_h, a_h) \phi(s_h, a_h)^\top]_{(ij)} \right| \leq \frac{C_3 d H \bar{\epsilon} \iota}{4}. \tag{42}$$

---

[15] We ignore the extreme case where $H$ is super large for simplicity. When $H$ is very large, we can simply construct $\Pi_{\epsilon/H}^{exp}$ instead and the proof is identical.

*Proof of Lemma F.8.* We have

$$
\begin{aligned}
LHS \leq &c' \sqrt{d^2 \log(\frac{3Hd}{\epsilon\delta}) + d\log(\frac{N}{\delta}) + 2\log(d)} \cdot \mathbb{E}_\pi \sum_{\widetilde{h}=1}^{h-1} \|\phi(s_{\widetilde{h}}, a_{\widetilde{h}})\|_{(\Lambda_{\widetilde{h}}^{h-1})^{-1}} \\
\leq &c_E \cdot d\log(\frac{C_2 dH}{C_1 \epsilon\delta}) \cdot H\bar{\epsilon} \\
\leq &\frac{C_3 dH\bar{\epsilon}\iota}{4}.
\end{aligned}
\tag{43}
$$

The first inequality holds because Lemma E.5. The second inequality results from Lemma F.4 and our induction condition F.7. The last inequality is due to our choice of $C_3$ (33). $\qquad\square$

Now we can bound $\left\|\widehat{\mathbb{E}}_\pi[\phi(s_h, a_h)\phi(s_h, a_h)^\top] - \mathbb{E}_\pi[\phi(s_h, a_h)\phi(s_h, a_h)^\top]\right\|_2$ by the following lemma.

**Lemma F.9** ($\ell_2$ norm bound). *With probability $1 - \delta$, for all $\pi \in \Pi_{\frac{\epsilon}{3}}^{exp}$, it holds that*

$$
\left\|\widehat{\mathbb{E}}_\pi[\phi(s_h, a_h)\phi(s_h, a_h)^\top] - \mathbb{E}_\pi[\phi(s_h, a_h)\phi(s_h, a_h)^\top]\right\|_2 \leq \frac{C_3 d^2 H\bar{\epsilon}\iota}{4}.
\tag{44}
$$

*Proof of Lemma F.9.* The inequality results from Lemma F.8 and the fact that for any $X \in \mathbb{R}^{d\times d}$,

$$
\|X\|_2 \leq \|X\|_F.
\tag{45}
$$

$\qquad\square$

Note that the conclusion also holds for all $\pi \in \Delta(\Pi_{\frac{\epsilon}{3}}^{exp})$ due to our discussion in Remark E.7.

According to our condition that $\epsilon < \frac{H(\lambda^\star)^2}{C_4 d^{7/2}\log(1/\lambda^\star)} = \frac{H(\lambda^\star)^2}{80 C_1 C_3 d^{7/2}\log(1/\lambda^\star)}$ and (41), we have

$$
\lambda_{\min}(\mathbb{E}_{\bar{\pi}_h^\star}\phi_h\phi_h^\top) \geq \frac{(\lambda^\star)^2}{64 d^2 \log(1/\lambda^\star)} \geq \frac{5 C_1 C_3 d^{3/2}\epsilon}{4H} = \frac{5 C_3 d^2 H\bar{\epsilon}\iota}{4}.
\tag{46}
$$

Therefore, under the high probability case in Lemma F.9, due to Weyl's inequality,

$$
\lambda_{\min}(\widehat{\mathbb{E}}_{\bar{\pi}_h^\star}\phi_h\phi_h^\top) \geq C_3 d^2 H\bar{\epsilon}\iota.
\tag{47}
$$

We have (47) implies that $\bar{\pi}_h^\star$ is a feasible solution of the optimization problem (1) and therefore,

$$
\max_{\widehat{\pi}\in\Pi_{\frac{\epsilon}{3}}^{exp}} \widehat{\mathbb{E}}_{\widehat{\pi}}\left[\phi(s_h, a_h)^\top (N\cdot\widehat{\Sigma}_{\pi_h})^{-1}\phi(s_h, a_h)\right] \leq \max_{\widehat{\pi}\in\Pi_{\frac{\epsilon}{3}}^{exp}} \widehat{\mathbb{E}}_{\widehat{\pi}}\left[\phi(s_h, a_h)^\top (N\cdot\widehat{\Sigma}_{\bar{\pi}_h^\star})^{-1}\phi(s_h, a_h)\right],
\tag{48}
$$

where $\pi_h$ is the policy we apply to explore layer $h$ and $\lambda_{\min}(\widehat{\Sigma}_{\pi_h}) \geq C_3 d^2 H\bar{\epsilon}\iota$.

### F.5.2 Error bound for the second estimator

We consider the upper bound of

$$
\left|\widehat{\mathbb{E}}_{\widehat{\pi}}\left[\phi(s_h, a_h)^\top (N\cdot\widehat{\Sigma}_\pi)^{-1}\phi(s_h, a_h)\right] - \mathbb{E}_{\widehat{\pi}}\left[\phi(s_h, a_h)^\top (N\cdot\widehat{\Sigma}_\pi)^{-1}\phi(s_h, a_h)\right]\right|.
$$

Recall that $\widehat{\mathbb{E}}_{\widehat{\pi}}\left[\phi(s_h, a_h)^\top (N\cdot\widehat{\Sigma}_\pi)^{-1}\phi(s_h, a_h)\right]$ is calculated by calling Algorithm 4 with $A = \frac{\bar{\epsilon}}{C_2 d^3 H\iota^2}$. Note that we only need to consider the case where $\widehat{\pi} \in \Pi_{\frac{\epsilon}{3}}^{exp}$, $\pi \in \Delta(\Pi_{\frac{\epsilon}{3}}^{exp})$ and $\lambda_{\min}(\widehat{\Sigma}_\pi) \geq C_3 d^2 H\bar{\epsilon}\iota$.

**Lemma F.10.** *With probability $1 - \delta$, for all $\widehat{\pi} \in \Pi_{\frac{\epsilon}{3}}^{exp}$ and all $\pi \in \Delta(\Pi_{\frac{\epsilon}{3}}^{exp})$ such that $\lambda_{\min}(\widehat{\Sigma}_\pi) \geq C_3 d^2 H\bar{\epsilon}\iota$, it holds that:*

$$
\left|\widehat{\mathbb{E}}_{\widehat{\pi}}\left[\phi(s_h, a_h)^\top (N\cdot\widehat{\Sigma}_\pi)^{-1}\phi(s_h, a_h)\right] - \mathbb{E}_{\widehat{\pi}}\left[\phi(s_h, a_h)^\top (N\cdot\widehat{\Sigma}_\pi)^{-1}\phi(s_h, a_h)\right]\right| \leq \frac{\bar{\epsilon}^2}{2d^2}.
\tag{49}
$$

*Proof of Lemma F.10.* We have

$$
\begin{aligned}
LHS \leq & c_E \cdot \frac{\bar{\epsilon}}{C_2 d^2 H \iota} \log(\frac{C_2 dH}{C_1 \epsilon \delta}) \cdot \mathbb{E}_{\widehat{\pi}} \sum_{\widetilde{h}=1}^{h-1} \|\phi(s_{\widetilde{h}}, a_{\widetilde{h}})\|_{(\Lambda_{\widetilde{h}}^{h-1})^{-1}} \\
\leq & \frac{\bar{\epsilon}}{2d^2 H} \cdot H\bar{\epsilon} = \frac{\bar{\epsilon}^2}{2d^2}.
\end{aligned}
\tag{50}
$$

The first inequality results from Lemma E.5 and Lemma F.5. The second inequality holds since our choice of $C_2$ (32) and induction condition F.7. □

**Remark F.11.** *We have with probability $1 - \delta$ (under the high probability case in Lemma F.10), due to the property of $\max\{\cdot\}$, for all $\pi \in \Delta(\Pi_{\frac{\epsilon}{3}}^{exp})$ such that $\lambda_{\min}(\widehat{\Sigma}_\pi) \geq C_3 d^2 H \bar{\epsilon} \iota$, it holds that:*

$$
\left| \max_{\widehat{\pi} \in \Pi_{\frac{\epsilon}{3}}^{exp}} \widehat{\mathbb{E}}_{\widehat{\pi}} \left[ \phi(s_h, a_h)^\top (N \cdot \widehat{\Sigma}_\pi)^{-1} \phi(s_h, a_h) \right] - \max_{\widehat{\pi} \in \Pi_{\frac{\epsilon}{3}}^{exp}} \mathbb{E}_{\widehat{\pi}} \left[ \phi(s_h, a_h)^\top (N \cdot \widehat{\Sigma}_\pi)^{-1} \phi(s_h, a_h) \right] \right| \leq \frac{\bar{\epsilon}^2}{2d^2}.
\tag{51}
$$

## F.6 Main proof

With all preparations ready, we are ready to prove the main theorem. We assume the high probability cases in Lemma F.8 (which implies Lemma F.9) and Lemma F.10 hold. First of all, we have:

$$
\begin{aligned}
& \max_{\widehat{\pi} \in \Pi_{\frac{\epsilon}{3}}^{exp}} \widehat{\mathbb{E}}_{\widehat{\pi}} \left[ \phi(s_h, a_h)^\top (N \cdot \widehat{\Sigma}_{\bar{\pi}_h^\star})^{-1} \phi(s_h, a_h) \right] \\
& \leq \max_{\widehat{\pi} \in \Pi_{\frac{\epsilon}{3}}^{exp}} \mathbb{E}_{\widehat{\pi}} \left[ \phi(s_h, a_h)^\top (N \cdot \widehat{\Sigma}_{\bar{\pi}_h^\star})^{-1} \phi(s_h, a_h) \right] + \frac{\bar{\epsilon}^2}{2d^2} \\
& \leq \max_{\widehat{\pi} \in \Pi_{\frac{\epsilon}{3}}^{exp}} \mathbb{E}_{\widehat{\pi}} \left[ \phi(s_h, a_h)^\top (N \cdot \widehat{\Sigma}_{\bar{\pi}_h^\star})^{-1} \phi(s_h, a_h) \right] + \frac{\bar{\epsilon}^2}{8} \\
& \leq \max_{\widehat{\pi} \in \Pi_{\frac{\epsilon}{3}}^{exp}} \mathbb{E}_{\widehat{\pi}} \left[ \phi(s_h, a_h)^\top (\frac{4N}{5} \cdot \Sigma_{\bar{\pi}_h^\star})^{-1} \phi(s_h, a_h) \right] + \frac{\bar{\epsilon}^2}{8} \\
& \leq \frac{5d}{4N} + \frac{\bar{\epsilon}^2}{8} \leq \frac{3\bar{\epsilon}^2}{8}.
\end{aligned}
\tag{52}
$$

The first inequality holds because of Lemma F.10 (and Remark F.11). The second inequality is because under meaningful case, $d \geq 2$. The third inequality holds since under the high probability case in Lemma F.9, $\frac{\Sigma_{\bar{\pi}_h^\star}}{5} \succcurlyeq \frac{C_3 d^2 H \bar{\epsilon} \iota}{4} I_d \succcurlyeq \Sigma_{\bar{\pi}_h^\star} - \widehat{\Sigma}_{\bar{\pi}_h^\star}$ can imply $\widehat{\Sigma}_{\bar{\pi}_h^\star} \succcurlyeq \frac{4}{5} \Sigma_{\bar{\pi}_h^\star}$, and thus $(\widehat{\Sigma}_{\bar{\pi}_h^\star})^{-1} \preccurlyeq (\frac{4}{5} \Sigma_{\bar{\pi}_h^\star})^{-1}$.[16] The forth inequality is due to the definition of $\bar{\pi}_h^\star$ and Theorem B.1. The last inequality holds since our choice of $N$ and $C_2$.

Combining (52) and (48), we have

$$
\frac{3\bar{\epsilon}^2}{8} \geq \max_{\widehat{\pi} \in \Pi_{\frac{\epsilon}{3}}^{exp}} \widehat{\mathbb{E}}_{\widehat{\pi}} \left[ \phi(s_h, a_h)^\top (N \cdot \widehat{\Sigma}_{\pi_h})^{-1} \phi(s_h, a_h) \right].
\tag{53}
$$

According to Lemma F.10, Remark F.11 and the fact that $\lambda_{\min}(\widehat{\Sigma}_{\pi_h}) \geq C_3 d^2 H \bar{\epsilon} \iota$. It holds that

$$
\frac{3\bar{\epsilon}^2}{8} \geq \max_{\widehat{\pi} \in \Pi_{\frac{\epsilon}{3}}^{exp}} \mathbb{E}_{\widehat{\pi}} \left[ \phi(s_h, a_h)^\top (N \cdot \widehat{\Sigma}_{\pi_h})^{-1} \phi(s_h, a_h) \right] - \frac{\bar{\epsilon}^2}{8}.
\tag{54}
$$

Or equivalently, $\frac{\bar{\epsilon}^2}{2} \geq \max_{\widehat{\pi} \in \Pi_{\frac{\epsilon}{3}}^{exp}} \mathbb{E}_{\widehat{\pi}} \left[ \phi(s_h, a_h)^\top (N \cdot \widehat{\Sigma}_{\pi_h})^{-1} \phi(s_h, a_h) \right]$.

---

[16]Note that all matrices here are symmetric and positive definite.

Suppose after applying policy $\pi_h$ for $N$ episodes, the data we collect[17] is $\{s_h^i, a_h^i\}_{i \in [N]}$. Assume $\bar{\Lambda}_h = I + \sum_{i=1}^{N} \phi(s_h^i, a_h^i)\phi(s_h^i, a_h^i)^{\top}$, we now consider the relationship between $\bar{\Lambda}_h$ and $\widehat{\Sigma}_{\pi_h}$.

First, according to Lemma F.9, we have:

$$N \cdot \widehat{\Sigma}_{\pi_h} - N \cdot \Sigma_{\pi_h} \preccurlyeq \frac{C_3 N d^2 H \bar{\epsilon} \iota}{4} \cdot I_d \preccurlyeq \frac{1}{4} N \cdot \widehat{\Sigma}_{\pi_h}. \tag{55}$$

Besides, due to Lemma F.1 (with $C = 1$), with probability $1 - \delta$,

$$N \cdot \Sigma_{\pi_h} - \bar{\Lambda}_h \preccurlyeq 4\sqrt{2}\sqrt{N\iota} \cdot I_d \preccurlyeq \frac{C_3 N d^2 H \bar{\epsilon} \iota}{4} \cdot I_d \preccurlyeq \frac{1}{4} N \cdot \widehat{\Sigma}_{\pi_h}. \tag{56}$$

Combining (55) and (56), we have with probability $1 - \delta$,

$$N \cdot \widehat{\Sigma}_{\pi_h} - \bar{\Lambda}_h \preccurlyeq \frac{1}{2} N \cdot \widehat{\Sigma}_{\pi_h}, \tag{57}$$

or equivalently,

$$(N \cdot \widehat{\Sigma}_{\pi_h})^{-1} \succcurlyeq (2\bar{\Lambda}_h)^{-1}. \tag{58}$$

Plugging (58) into (54), we have with probability $1 - \delta$,

$$
\begin{aligned}
\frac{\bar{\epsilon}^2}{2} &\geq \max_{\widehat{\pi} \in \Pi_{\frac{\epsilon}{3}}^{exp}} \mathbb{E}_{\widehat{\pi}} \left[ \phi(s_h, a_h)^{\top} (N \cdot \widehat{\Sigma}_{\pi_h})^{-1} \phi(s_h, a_h) \right] \\
&\geq \max_{\widehat{\pi} \in \Pi_{\frac{\epsilon}{3}}^{exp}} \mathbb{E}_{\widehat{\pi}} \left[ \phi(s_h, a_h)^{\top} (2\bar{\Lambda}_h)^{-1} \phi(s_h, a_h) \right] \\
&\geq \frac{1}{2} \left( \max_{\widehat{\pi} \in \Pi_{\frac{\epsilon}{3}}^{exp}} \mathbb{E}_{\widehat{\pi}} \sqrt{\phi(s_h, a_h)^{\top} (\bar{\Lambda}_h)^{-1} \phi(s_h, a_h)} \right)^2,
\end{aligned}
\tag{59}
$$

where the last inequality follows Cauchy-Schwarz inequality.

Recall that after the exploration of layer $h$, $\Lambda_h^h$ in (35) uses all previous data up to the $h$-th deployment, which implies that $\Lambda_h^h \succcurlyeq \bar{\Lambda}_h$ and $(\Lambda_h^h)^{-1} \preccurlyeq (\bar{\Lambda}_h)^{-1}$. Therefore, with probability $1 - \delta$,

$$\bar{\epsilon} \geq \max_{\widehat{\pi} \in \Pi_{\frac{\epsilon}{3}}^{exp}} \mathbb{E}_{\widehat{\pi}} \sqrt{\phi(s_h, a_h)^{\top} (\bar{\Lambda}_h)^{-1} \phi(s_h, a_h)} \geq \max_{\widehat{\pi} \in \Pi_{\frac{\epsilon}{3}}^{exp}} \mathbb{E}_{\widehat{\pi}} \sqrt{\phi(s_h, a_h)^{\top} (\Lambda_h^h)^{-1} \phi(s_h, a_h)}, \tag{60}$$

which implies that the induction process holds.

Recall that after the whole exploration process for all $H$ layers, the dataset $\mathcal{D} = \{s_h^n, a_h^n\}_{h,n \in [H] \times [HN]}$ and $\Lambda_h = I + \sum_{n=1}^{HN} \phi_h^n (\phi_h^n)^{\top}$ for all $h \in [H]$. Due to induction, we have with probability $1 - H\delta$,

$$\max_{\pi \in \Pi_{\frac{\epsilon}{3}}^{exp}} \mathbb{E}_{\pi} \left[ \sum_{h=1}^{H} \sqrt{\phi(s_h, a_h)^{\top} (\Lambda_h)^{-1} \phi(s_h, a_h)} \right] \leq H\bar{\epsilon}. \tag{61}$$

In addition, according to Lemma C.7, $\Pi_{\frac{\epsilon}{3}}^{eval} \subseteq \Pi_{\frac{\epsilon}{3}}^{exp}$, we have

$$\max_{\pi \in \Pi_{\frac{\epsilon}{3}}^{eval}} \mathbb{E}_{\pi} \left[ \sum_{h=1}^{H} \sqrt{\phi(s_h, a_h)^{\top} (\Lambda_h)^{-1} \phi(s_h, a_h)} \right] \leq H\bar{\epsilon}. \tag{62}$$

Given (62), we are ready to prove the final result. Recall that the output of Algorithm 3 (with input $\pi$ and $r$) is $\widehat{V}^{\pi}(r)$. With probability $1 - \delta$, for all feasible linear reward function $r$, for all $\pi \in \Pi_{\frac{\epsilon}{3}}^{eval}$, it holds that

$$
\begin{aligned}
|\widehat{V}^{\pi}(r) - V^{\pi}(r)| &\leq c' H \sqrt{d} \cdot \sqrt{\log(\frac{3Hd}{\epsilon\delta}) + \log(\frac{N}{\delta})} \cdot \mathbb{E}_{\pi} \sum_{h=1}^{H} \|\phi(s_h, a_h)\|_{\Lambda_h^{-1}} \\
&\leq c_D H \sqrt{d} \cdot \log(\frac{C_2 dH}{C_1 \epsilon \delta}) \cdot H\bar{\epsilon} \\
&\leq \frac{1}{3C_1} H \sqrt{d}\iota \cdot H\bar{\epsilon} = \frac{\epsilon}{3},
\end{aligned}
\tag{63}
$$

---

[17]We only consider the data from layer $h$.

where the first inequality holds due to Lemma D.6. The second inequality is because of Lemma F.2 and (62). The third inequality holds since our choice of $C_1$ (31). The last equation results from our definition that $\bar{\epsilon} = \frac{C_1\epsilon}{H^2\sqrt{d\iota}}$.

Suppose $\widetilde{\pi}(r) = \arg\max_{\pi\in\Pi^{eval}_{\frac{\epsilon}{3}}} V^\pi(r)$. Since our output policy $\widehat{\pi}(r)$ is the greedy policy with respect to $\widehat{V}^\pi(r)$, we have

$$
\begin{aligned}
V^{\widetilde{\pi}(r)}(r) - V^{\widehat{\pi}(r)}(r) \leq & V^{\widetilde{\pi}(r)}(r) - \widehat{V}^{\widetilde{\pi}(r)}(r) + \widehat{V}^{\widetilde{\pi}(r)}(r) - \widehat{V}^{\widehat{\pi}(r)}(r) + \widehat{V}^{\widehat{\pi}(r)}(r) - V^{\widehat{\pi}(r)}(r) \\
\leq & \frac{2\epsilon}{3}.
\end{aligned}
\tag{64}
$$

In addition, according to Lemma C.5, $V^\star(r) - V^{\widetilde{\pi}(r)}(r) \leq \frac{\epsilon}{3}$. Combining these two results, we have with probability $1 - \delta$, for all feasible linear reward function $r$,

$$
V^\star(r) - V^{\widehat{\pi}(r)}(r) \leq \epsilon.
\tag{65}
$$

Since the deployment complexity of Algorithm 1 is clearly bounded by $H$, the proof of Theorem 5.1 is completed.

# G  Comparisons on results and techniques

In this section, we compare our results with the closest related work [Huang et al., 2022]. We begin with comparison of the conditions.

**Comparison of conditions.** In Assumption 2.1, we assume that the linear MDP satisfies

$$
\lambda^\star = \min_{h\in[H]} \sup_\pi \lambda_{\min}(\mathbb{E}_\pi[\phi(s_h, a_h)\phi(s_h, a_h)^\top]) > 0.
$$

In comparison, Huang et al. [2022] assume that

$$
\nu_{\min} = \min_{h\in[H]} \min_{\|\theta\|=1} \max_\pi \sqrt{\mathbb{E}_\pi[(\phi_h^\top\theta)^2]} > 0.
$$

Overall these two assumptions are analogous reachability assumptions, while our assumption is slightly stronger since $\nu_{\min}^2$ is lower bounded by $\lambda^\star$.

**Dependence on reachability coefficient.** Our Algorithm 1 only takes $\epsilon$ as input and does not require the knowledge of $\lambda^\star$, while the theoretical guarantee in Theorem 5.1 requires additional condition that $\epsilon$ is small compared to $\lambda^\star$. For $\epsilon$ larger than a problem-dependent threshold, the theoretical guarantee no longer holds. Such dependence is similar to the dependence on reachability coefficient $\nu_{\min}$ in Zanette et al. [2020b] where their algorithm also takes $\epsilon$ as input and requires $\epsilon$ to be small compared to $\nu_{\min}$. In comparison, Algorithm 2 in Huang et al. [2022] takes the reachability coefficient $\nu_{\min}$ as input, which is a stronger requirement than requiring $\epsilon$ to be small compared to $\lambda^\star$.

**Comparison of sample complexity bounds.** Our main improvement over Huang et al. [2022] is on the sample complexity bound in the small-$\epsilon$ regime. Comparing our asymptotic sample complexity bound $\widetilde{O}(\frac{d^2H^5}{\epsilon^2})$ with $\widetilde{O}(\frac{d^3H^5}{\epsilon^2\nu_{\min}^2})$ in Huang et al. [2022], our bound is better by a factor of $\frac{d}{\nu_{\min}^2}$, where $\nu_{\min}$ is always upper bounded by 1 and can be arbitrarily small (please see the illustration below). In the large-$\epsilon$ regime, the sample complexity bounds in both works look like $poly(d, H, \frac{1}{\lambda^\star})$ (or $poly(d, H, \frac{1}{\nu_{\min}})$), and such "Burn in" period is common in optimal experiment design based works [Wagenmaker and Jamieson, 2022].

**Illustration of $\nu_{\min}$.** In this part, we construct some examples to show what $\nu_{\min}$ will be like. First, consider the following simple example where the linear MDP 1 is defined as:

1. The linear MDP is a tabular MDP with only one action and several states ($A = 1, S > 1$).

2. The features are canonical basis [Jin et al., 2020b] and thus $d = S$.

3. The transition from any $(s, a) \in \mathcal{S} \times \mathcal{A}$ at any time step $h \in [H]$ is uniformly random.

Therefore, under linear MDP 1, both $\nu_{\min}^2$ in Huang et al. [2022] and our $\lambda^\star$ are $\frac{1}{d}$ and our improvement on sample complexity is a factor of $d^2$. Generally speaking, this example has a relatively large $\nu_{\min}$, and there are various examples with even smaller $\nu_{\min}$. Next, we construct the linear MDP 2 that is similar to the linear MDP 1 but does not have uniform transition kernel:

1. The linear MDP is a tabular MDP with only one action and several states ($A = 1$, $S > 1$).

2. The features are canonical basis [Jin et al., 2020b] and thus $d = S$.

3. The transitions from any $(s, a) \in \mathcal{S} \times \mathcal{A}$ at any time step $h \in [H]$ are the same and satisfies $\min_{s' \in \mathcal{S}} P_h(s'|s, a) = p_{\min}$.

Therefore, under linear MDP 2, both $\nu_{\min}^2$ in Huang et al. [2022] and our $\lambda^\star$ are $p_{\min}$ ($p_{\min} \leq \frac{1}{d}$) and our improvement on sample complexity is a factor of $d/p_{\min}$ which is always larger than $d^2$ and can be much larger. In the worst case, according to the condition ($\epsilon < \nu_{\min}^8$) for the asymptotic sample complexity in Huang et al. [2022] to dominate, $p_{\min} = \nu_{\min}^2$ can be as small as $\epsilon^{1/4}$, and the sample complexity in Huang et al. [2022] is $\widetilde{O}(\frac{1}{\epsilon^{2.25}})$, which does not have optimal dependence on $\epsilon$. In conclusion, our improvement on sample complexity is at least a factor of $d$ and can be much more significant under various circumstances.

**Technique comparison.** We discuss why we can get rid of the $\frac{d}{\nu_{\min}^2}$ dependence in Huang et al. [2022]. First, instead of minimizing $\max_\pi \mathbb{E}_\pi \|\phi_h\|_{\Lambda_h^{-1}}$, we only minimize the smaller $\max_{\pi \in \Pi_{\epsilon/3}^{exp}} \mathbb{E}_\pi \|\phi_h\|_{\Lambda_h^{-1}}$, where the maximum is taken over our explorative policy set. Therefore, our approximation of generalized G-optimal design helps save the factor of $1/\nu_{\min}^2$. In addition, note that in Lemma 6.3, the dependence on $d$ is only $\sqrt{d}$, this is because we estimate the value functions (w.r.t $\pi$ and $r$) instead of adding optimism and using LSVI. Compared to the log-covering number $\widetilde{O}(d^2)$ of the bonus term $\sqrt{\phi_h^\top \Lambda^{-1} \phi_h}$, our covering of (policy $\pi_h \in \Pi_{\epsilon/3,h}^{eval}$, linear reward $r_h$) has log-covering number $\widetilde{O}(d)$.

# H    Proof for Section 7

## H.1    Application to tabular MDP

Recall that the tabular MDP has discrete state-action space with $|\mathcal{S}| = S$, $|\mathcal{A}| = A$. We transfer our Assumption 2.1 to its counterpart under tabular MDP, and assume it holds.

**Assumption H.1.** *Define $d_h^\pi(\cdot, \cdot)$ to be the occupancy measure, i.e. $d_h^\pi(s, a) = \mathbb{P}_\pi(s_h = s, a_h = a)$. Let $d_m = \min_h \sup_\pi \min_{s,a} d_h^\pi(s, a)$, we assume that $d_m > 0$.*

**Theorem H.2.** *We select $\bar{\epsilon} = \frac{C_1 \epsilon}{H^2 \sqrt{S}\iota}$, $\Pi^{exp} = \Pi^{eval} = \Pi_0 = \{$all deterministic policies$\}$ and $N = \frac{C_2 S A \iota}{\bar{\epsilon}^2} = \frac{C_2 S^2 A H^4 \iota^3}{C_1^2 \epsilon^2}$ in Algorithm 1 and 2. The optimization problem is replaced by*

$$\pi_h = \underset{\pi \in \Delta(\Pi_0) \, s.t. \, \forall \, (s,a), \, \widehat{d}_h^\pi(s,a) \geq C_3 H \sqrt{S} \bar{\epsilon}\iota}{\operatorname{argmin}} \max_{\widehat{\pi} \in \Pi_0} \sum_{s,a} \frac{\widehat{d}_h^{\widehat{\pi}}(s, a)}{\widehat{d}_h^\pi(s, a)}, \tag{66}$$

*where $\widehat{d}_h^\pi(s, a)$ is estimated through applying Algorithm 4. Suppose $\epsilon \leq \frac{H d_m}{C_4 S A}$, with probability $1 - \delta$, for any reward function $r$, Algorithm 2 returns a policy that is $\epsilon$-optimal with respect to $r$. In addition, the deployment complexity of Algorithm 1 is $H$ while the number of trajectories is $\widetilde{O}(\frac{S^2 A H^5}{\epsilon^2})$.*

*Proof of Theorem H.2.* Since the proof is quite similar to the proof of Theorem 5.1, we sketch the proof and highlight the difference to the linear MDP setting while ignoring details.

Suppose after the $h$-th deployment, the visitation number of $(\widetilde{h}, s, a)$ is $N_{\widetilde{h}}^h(s, a)$. Then our induction condition becomes after the $(h - 1)$-th deployment, $\max_\pi \left[ \sum_{\widetilde{h}=1}^{h-1} \sum_{s,a} \frac{d_h^\pi(s,a)}{\sqrt{N_{\widetilde{h}}^{h-1}(s,a)}} \right] \leq (h - 1)\bar{\epsilon}$.

We base on this condition and prove that with high probability, $\max_\pi \left[ \sum_{s,a} \frac{d_h^\pi(s,a)}{\sqrt{N_h^h(s,a)}} \right] \leq \bar{\epsilon}$.

First, under tabular MDP, Algorithm 4 is equivalent to value iteration based on empirical transition kernel. Therefore, due to standard methods like simulation lemma, we have with high probability, for any $\pi \in \Pi_0$ and reward $r$ with upper bound $A$ (the $V_{\widetilde{h}}$ function is the one we derive in Algorithm 4),

$$
\begin{aligned}
|\widehat{\mathbb{E}}_\pi r(s_h, a_h) - \mathbb{E}_\pi r(s_h, a_h)| \leq & \mathbb{E}_\pi \sum_{\widetilde{h}=1}^{h-1} \left| \left( \widehat{P}_{\widetilde{h}} - P_{\widetilde{h}} \right) \cdot V_{\widetilde{h}+1}(s_{\widetilde{h}}, a_{\widetilde{h}}) \right| \\
\leq & \mathbb{E}_\pi \sum_{\widetilde{h}=1}^{h-1} A \cdot \left\| \widehat{P}_{\widetilde{h}} - P_{\widetilde{h}} \right\|_1 \\
\leq & \widetilde{O} \left( A\sqrt{S} \cdot \mathbb{E}_\pi \sum_{\widetilde{h}=1}^{h-1} \sqrt{\frac{1}{N_{\widetilde{h}}^{h-1}(s_{\widetilde{h}}, a_{\widetilde{h}})}} \right) \\
\leq & \widetilde{O} \left( A\sqrt{S} \cdot \sum_{\widetilde{h}=1}^{h-1} \sum_{s,a} \frac{d_{\widetilde{h}}^\pi(s,a)}{\sqrt{N_{\widetilde{h}}^{h-1}(s,a)}} \right) \\
\leq & A\sqrt{S} \cdot H\bar{\epsilon}.
\end{aligned}
\tag{67}
$$

Now we prove that our condition about $\epsilon$ is enough. Note that with high probability, for all policy $\pi \in \Pi_0$ and $s, a$, the estimation error of $\widehat{d}_h^\pi(s,a)$ is bounded by $\sqrt{S} \cdot H\bar{\epsilon}$. As a result, the estimation error can be ignored compared to $d_h^{\pi_h}(s,a)$ or $d_h^{\pi_h^\star}(s,a)$. With identical proof to Section F.6, we have the induction still holds.

From the induction, suppose $N_h(s,a)$ is the final visitation number of $(h, s, a)$, we have $\max_\pi \left[ \sum_{h=1}^{H} \sum_{s,a} \frac{d_h^\pi(s,a)}{\sqrt{N_h(s,a)}} \right] \leq H\bar{\epsilon}$. Using identical proof to (67), we have with high probability, for all $\pi \in \Pi_0$ and $r$,

$$
|\widehat{V}^\pi(r) - V^\pi(r)| \leq \widetilde{O}(H\sqrt{S} \cdot H\bar{\epsilon}) \leq \frac{\epsilon}{2}.
\tag{68}
$$

Since $\Pi_0$ contains the optimal policy, our output policy is $\epsilon$-optimal. $\square$

## H.2 Proof of lower bounds

For regret minimization, we assume the number of episodes is $K$ while the number of steps is $T := KH$.

**Theorem H.3** (Restate Theorem 7.2). *For any algorithm with the optimal $\widetilde{O}(\sqrt{poly(d, H)T})$ regret bound, the switching cost is at least $\Omega(dH \log \log T)$.*

*Proof of Theorem H.3.* We first construct a linear MDP with two states, the initial state $s_1$ and the absorbing state $s_2$.

For absorbing state $s_2$, the choice of action is only $a_0$, while for initial state $s_1$, the choice of actions is $\{a_1, a_2, \cdots, a_{d-1}\}$. Then we define the feature map:

$$
\phi(s_2, a_0) = (1, 0, 0, \cdots, 0), \quad \phi(s_1, a_i) = (0, \cdots, 0, 1, 0, \cdots),
$$

where for $s_1, a_i$ ($i \in [d-1]$), the $(i+1)$-th element is 1 while all other elements are 0. We now define the measure $\mu_h$ and reward vector $\theta_h$ as:

$$
\mu_h(s_1) = (0, 1, 0, 0, \cdots, 0), \quad \mu_h(s_2) = (1, 0, 1, 1, \cdots, 1), \quad \forall h \in [H].
$$

$$
\theta_h = (0, 0, r_{h,2}, \cdots, r_{h,d-1}), \quad \text{where } r_{h,i}\text{'s are unknown non-zero values.}
$$

Combining these definitions, we have: $P_h(s_2|s_2, a_0) = 1$, $r_h(s_2, a_0) = 0$, $P_h(s_1|s_1, a_1) = 1$, $r_h(s_1, a_1) = 0$ for all $h \in [H]$. Besides, $P_h(s_2|s_1, a_i) = 1$, $r_h(s_1, a_i) = r_{h,i}$ for all $h \in [H], i \geq 2$.

Therefore, for any deterministic policy, the only possible case is that the agent takes action $a_1$ and stays at $s_1$ for the first $h - 1$ steps, then at step $h$ the agent takes action $a_i$ ($i \geq 2$) and transitions to $s_2$ with reward $r_{h,i}$, later the agent always stays at $s_2$ with no more reward. For this trajectory,

the total reward will be $r_{h,i}$. Also, for any deterministic policy, the trajectory is fixed, like pulling an "arm" in multi-armed bandits setting. Note that the total number of such "arms" with non-zero unknown reward is at least $(d-2)H$. Even if the transition kernel is known to the agent, this linear MDP is still as difficult as a multi-armed bandits problem with $\Omega(dH)$ arms. Together will Lemma H.4 below, the proof is complete. $\qquad\square$

**Lemma H.4** (Theorem 2 in [Simchi-Levi and Xu, 2019]). *Under the $K$-armed bandits problem, there exists an absolute constant $C > 0$ such that for all $K > 1, S \geq 0, T \geq 2K$ and for all policy $\pi$ with switching budget $S$, the regret satisfies*

$$R^\pi(K,T) \geq \frac{C}{\log T} \cdot K^{1-\frac{1}{2-2^{-q(S,K)-1}}} T^{\frac{1}{2-2^{-q(S,K)-1}}},$$

*where $q(S,K) = \lfloor \frac{S-1}{K-1} \rfloor$. This further implies that $\Omega(K \log \log T)$ switches are necessary for achieving $\widetilde{O}(\sqrt{T})$ regret bound.*

**Theorem H.5** (Restate Theorem 7.3). *For any algorithm with the optimal $\widetilde{O}(\sqrt{poly(d,H)T})$ regret bound, the number of batches is at least $\Omega(\frac{H}{\log_d T} + \log \log T)$.*

*Proof of Theorem H.5.* Corollary 2 of Gao et al. [2019] proved that under multi-armed bandits problem, for any algorithm with optimal $\widetilde{O}(\sqrt{T})$ regret bound, the number of batches is at least $\Omega(\log \log T)$. In the proof of Theorem H.3, we show that linear MDP can be at least as difficult as a multi-armed bandits problem, which means the $\Omega(\log \log T)$ lower bound on batches also applies to linear MDP.

In addition, Theorem B.3 in Huang et al. [2022] stated an $\Omega(\frac{H}{\log_d NH})$ lower bound for deployment complexity for any algorithm with PAC guarantee. Note that one deployment of arbitrary policy is equivalent to one batch. Suppose we can design an algorithm to get $\widetilde{O}(\sqrt{T})$ regret within $K$ episodes and $M$ batches, then we are able to identify near-optimal policy in $M$ deployments while each deployment is allowed to collect $K$ trajectories. Therefore, we have $M \geq \Omega(\frac{H}{\log_d T})$.

Combining these two results, the proof is complete. $\qquad\square$

