# OpenReview forum: "Near-Optimal Deployment Efficiency in Reward-Free Reinforcement Learning with Linear Function Approximation"
_NeurIPS.cc/2022/Workshop/Offline_RL — Offline RL Workshop NeurIPS 2022_

### Official Review · Reviewer_QTzD · 2022-10-08

**Rating:** 7
**Confidence:** 4

**Review:**

This paper studies the deployment efficient setting in linear MDP, and proposes algorithms achieving better sample complexity than previous methods and also gets rid of the knowledge of reachability coefficient. This is a nice and solid paper and I believe the contribution is important.

The only (small) concern is that whether this paper is close to the offline RL. But considering that the algorithms in the paper also involve how to collect and use offline data, I think it is appropriate to have this paper appear in this workshop.

---

### Official Review · Reviewer_X8uD · 2022-10-17
**Sufficient content for workshop but seems out of scope?**

**Rating:** 6
**Confidence:** 3

**Review:**

**Summary**: This paper studies the problem of deployment (i.e. switching cost) efficient RL with linear function approximation under a reward-free exploration setting. This work generalizes the result in Qiao et al 2022 from a tabular setting to a linear setting. In terms of the guarantee, they obtain the optimal O(H) deployment cost and a $d^2 H^5 / \epsilon^2$ sample complexity. In terms of design, they chose layer-by-layer exploration over doubling design to obtain slow adaptivity, which I think is more intuitive and interesting. This design somewhat resembles epoch-based learning for RL with low switching costs. To be able to collect information data with only $H$ policies, they need to find a policy that is explorative in all directions of the feature at each layer $h$ (Eq (1) in Algo. 1).

**Novelty.** In terms of result novelty (a.k.a. significance for me), this work seems to be the first to obtain optimal deployment cost and optimal dependence on $d$ and $\epsilon$ in the model-free setting with linear models. In terms of technical novelty, they claimed exploration-preserving policy discretization and a generalized G-optimal experiment design which I am not able to evaluate as I am not all well aware of the literature of this work. Either way, technical novelty should not be a big issue to me given the result novelty is interesting, which seems to be the case here.

**Clarity.** Putting a technical overview right after the intro and before actually explaining the algorithms and main results makes it really hard for me as someone's not working on this problem set to follow the paper. There are also some clarification questions.
- How should the constrained min-max optimization in Eq (1) in Algo. 1 be solved? It looks highly intractable to me.

**Relevance.** Note that this is not really an offline RL work and so I am not sure if it suits the workshop.